# Iterative Tool Usage Exploration for Multimodal Agents via Step-wise Preference Tuning

**Pengxiang Li**[1,2*] **Zhi Gao**[1,2,3,5*] **Bofei Zhang**[2] **Yapeng Mi**[2,4] **Xiaojian Ma**[2]
**Chenrui Shi**[1,2] **Tao Yuan**[2] **Yuwei Wu**[1,5✉] **Yunde Jia**[5,1] **Song-Chun Zhu**[2,3,6] **Qing Li**[2✉]

[1]Beijing Key Laboratory of Intelligent Information Technology,
School of Computer Science & Technology, Beijing Institute of Technology
[2]State Key Laboratory of General Artificial Intelligence, BIGAI
[3]State Key Laboratory of General Artificial Intelligence, Peking University [4]Harbin Institute of Technology
[5]Guangdong Laboratory of Machine Perception and Intelligent Computing, Shenzhen MSU-BIT University
[6]Department of Automation, Tsinghua University
https://sport-agents.github.io

## Abstract

Multimodal agents, which integrate a controller (*e.g.*, a vision language model) with external tools, have demonstrated remarkable capabilities in tackling complex multimodal tasks. Existing approaches for training these agents, both supervised fine-tuning and reinforcement learning, depend on extensive human-annotated task-answer pairs and tool trajectories. However, for complex multimodal tasks, such annotations are prohibitively expensive or impractical to obtain. In this paper, we propose an iterative tool usage exploration method for multimodal agents without any pre-collected data, namely SPORT, via step-wise preference optimization to refine the trajectories of tool usage. Our method enables multimodal agents to autonomously discover effective tool usage strategies through self-exploration and optimization, eliminating the bottleneck of human annotation. SPORT has four iterative components: task synthesis, step sampling, step verification, and preference tuning. We first synthesize multimodal tasks using language models. Then, we introduce a novel trajectory exploration scheme, where step sampling and step verification are executed alternately to solve synthesized tasks. In step sampling, the agent tries different tools and obtains corresponding results. In step verification, we employ a verifier to provide AI feedback to construct step-wise preference data. The data is subsequently used to update the controller for tool usage through preference tuning, producing a SPORT agent. By interacting with real environments, the SPORT agent gradually evolves into a more refined and capable system. Evaluation in the GTA and GAIA benchmarks shows that the SPORT agent achieves 6.41% and 3.64% improvements, underscoring the generalization and effectiveness introduced by our method.

## 1 Introduction

Leveraging large language models (LLMs) or vision-language models (VLMs) as controllers to call external tools (*e.g.*, web search, visual reasoning, file understanding, and object localization) has become a promising direction in building multimodal agents [48, 19, 14, 49, 30], achieving impressive performance for complex tasks [17, 37, 31, 13]. To enhance the planning and reasoning abilities of agents, existing studies focus on collecting tool usage trajectories to fine-tune the controller of an agent [21, 18], using human annotation or distillation from closed-source APIs. However, collecting high-quality tool usage data is labor-intensive and high-cost, and such pre-collected data may lead to biased distributions inconsistent with the target environments (such as task distributions and available tools), causing inferior generalization.

---

*Equal contribution.   ✉ Corresponding author.

39th Conference on Neural Information Processing Systems (NeurIPS 2025).

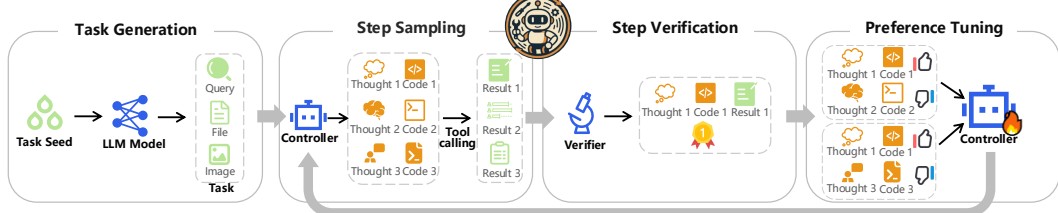

Figure 1: Pipeline of the proposed SPORT method, including four iterative components: task generation, step sampling, step verification, and preference tuning.

In this paper, we study whether multimodal agents can improve their tool usage capability via self-exploration without any pre-collection. We are inspired by existing research in LLMs and VLMs, which has shown impressive performance in self-instruction [53, 34], self-verification [40, 59], and self-learning [10, 25]. Based on the above observation, we expect that agents automatically generates tasks, searches for useful tools in solving synthetic tasks, evaluates the exploration by itself, and updates controllers using the exploration process. In this case, agents will improve the generalization of tool usage by interacting with environments.

To achieve this goal, we must address two key challenges in such tool usage exploration for complex multimodal tasks. (1) **Lack of off-the-shelf tasks and annotation.** There is no off-the-shelf dataset with ground truth annotations (answers and trajectories) for solving multimodal tasks. These tasks are usually domain-specific [51, 41], making it non-trivial to verify whether the outcome is correct. Solving these tasks requires a long trajectory to call diverse tools, and thus, it is challenging to produce an exactly correct trajectory and compare the quality among different trajectories. (2) **Low sampling efficiency and high cost.** The exploration process requires executing the sampled tools (*e.g.*, large language models, web search, and image generation), which results in both high monetary and computational costs, making it challenging to scale up.

To solve the above challenges, we propose SPORT, an iterative tool usage exploration method via step-wise preference optimization to refine trajectories of multimodal agents, as shown in Figure 1. SPORT operates through four iterative components: task synthesis, step sampling, step verification, and preference tuning. Firstly, we generate queries and multimodal files for task synthesis based on provided task seeds. Secondly, we introduce a new search scheme that samples step-level candidate actions (including the thoughts and codes) to call tools. Thirdly, we employ a multimodal verifier that, given the task context, step actions with execution results, provides AI-generated feedback to estimate step-level preferences. Finally, we perform step-wise preference tuning to refine the controller and obtain the SPORT agent, which is then used to guide tool sampling in the next iteration.

SPORT enables agents to autonomously generate tasks and explore tool usage trajectories, removing reliance on pre-collected datasets. Step-level verification is easier than trajectory-level evaluation for pre-trained LLMs, circumventing annotation difficulties. Furthermore, SPORT improves data utilization by extracting useful step-level preferences even from failed trajectories, enabling more effective learning with the same number of samples. These capabilities enable stable and scalable self-exploration, achieving better generalization for complex multimodal tool usage tasks.

We conduct experiments on the two multimodal reasoning benchmarks: GTA and GAIA, and results show that our SPORT agent outperforms the SFT agent by 6.41% and 3.64%, respectively. This indicates that our SPORT agent method leads to a more powerful reasoning and planning capability for tool usage by interacting with the environment.

In summary, our contributions are threefold. (1) We propose SPORT, a tool usage exploration framework for multimodal agents, providing a possible way for multimodal tool learning, leading to generalization without any annotation. (2) The obtained SPORT Agent achieves significant performance improvements compared with SFT agents on two popular benchmarks: GTA and GAIA. (3) We collect the explored preference tuning data into a dataset composed of 16K data, which is conducive to the subsequent research on tool usage and multimodal agents.

## 2 Related Work

### 2.1 Agent Tuning

Due to the disparity between the LLMs and the requirements of agents, agent tuning is necessary to adapt to practical tasks. Research for agent tuning could be divided into two categories: supervised

fine-tuning (SFT) and reinforcement learning (RL). SFT methods collect trajectory data via distillation from closed-source API (*e.g.*, GPT-4o) [18, 38, 60, 63] or human annotation [36, 9]. Then they use these collected data to tune the controller via SFT. However, the SFT methods suffer from huge costs and inferior generalization [46]. To solve this issue, researchers have paid attention to RL agent tuning methods that allow agents to interact with the environment and learn from the feedback. Some methods utilize the policy gradient technique [65] to update the controller with a reward model that is designed as a fine-tuned model [44, 62], environment feedback [2, 61], human-designed rules [42, 66], or tree search results [11]. Especially, some methods resort to the policy gradient method for tool learning [23, 15], where they use the prediction correctness as the reward. To simplify this procedure, the direct preference optimization (DPO) methods are applied to agent tuning [56], which construct step-level [43, 7] or trajectory-level [64, 47] preference data based on whole correct trajectories. Nevertheless, most existing agent tuning methods rely on answer or trajectory annotations that are difficult to obtain in multimodal tool usage tasks. In contrast, our tool usage exploration framework does not rely on any annotation via step-wise preference tuning.

## 2.2 Step-wise Preference Tuning

Preference tuning methods rely on paired data, which is not readily available for complex tasks with multi-step reasoning, making it non-trivial to determine which trajectory is better. Furthermore, for long trajectories, only an overall preference verification can not capture the relationships among steps and ignores the fine-grained preference between different steps. To overcome this problem, step-wise preference has been studied. STEP-DPO [26] and SCDPO [39] collect step-wise preference data by localizing error steps or disturbing the correct path. OREO [5] and SVPO [50] train value models for step-wise verification and inference guidance. SDPO [24] combines step-level, turn-level, and session-level preference data for full-grained optimization. The above methods mainly focus on code generation and math reasoning tasks that are easy to obtain correct trajectories to construct step-wise preference data. In contrast, this method focuses on multimodal agents for tool usage, where obtaining correct trajectories is challenging. Thus, our agent explores the tool usage by itself via an iterative manner, which uses designed AI feedback to construct step-wise preference data without any annotation.

## 2.3 Learning from AI Feedback

Using models to generate AI feedback for performance improvement has emerged as a critical paradigm [4]. Existing methods can be broadly divided into three categories. The first category adds AI feedback into prompts for in-context learning [40]. LLaVA-Cirtic [55] trains a model to provide multimodal AI feedback. VLM-F [32], VolCaNo [28], and Clarify [29] use AI feedback to address visual hallucinations. CLOVA [17] and CompAgent [54] refine prompts using AI feedback. The second category uses AI feedback to filter data for supervised fine-tuning, such as M-STAR [35] for visual mathematical reasoning and APIGen [38], MAT [18], and visualagentbench [36] for agent trajectories. The third category uses AI feedback for reinforcement learning, producing rewards for policy gradient optimization [27, 16, 45] or preference data for DPO [59]. Different from them, our AI feedback is well-designed for evaluating tool usage in complex scenarios of multimodal agents.

# 3 Method

## 3.1 Formulation

We opt for the framework of the ReAct agent [57] that performs step-by-step reasoning for tool usage. In each step, based on the input $x_i$, the agent outputs an action $a_i$ for tool calling.

$$a_i^\star = \arg\max \pi_\theta(a_i|x_i, T), \tag{1}$$

where $\pi_\theta$ is the controller (an VLM in our method) of agents with $\theta$ being the parameters, $x_i$ is composed of the task (including a query $Q$ in natural language and multimodal files $F$) and the history $h_i$ of previous steps, *i.e.*, $x_i = \{Q, F, h_i\}$. The action $a_i$ consists of the thought $t_i$ and code $c_i$ for tool calling, $a_i = \{t_i, c_i\}$. $T$ denotes available tools, and we follow the work [18] using the same toolkit. In this case, we further rewrite Eq. (1) as

$$t_i^\star, c_i^\star = \arg\max \pi_\theta(t_i, c_i|Q, F, h_i, T), \tag{2}$$

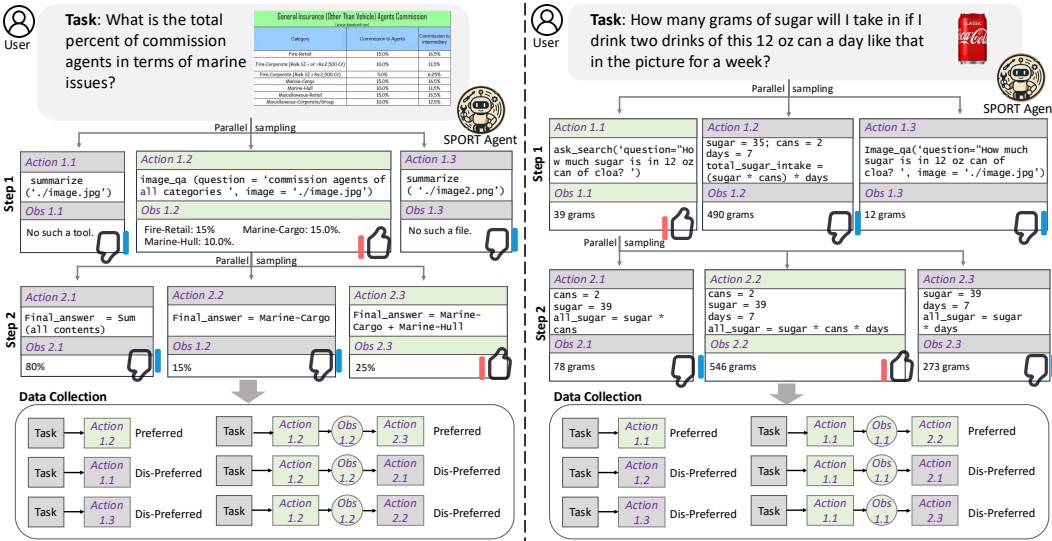

Figure 2: Demonstrations of the search scheme used in the SPORT method. Given a task, the agent samples potential actions for each step and verifies their qualities in an online manner. Then, we construct the step-wise preference data based on such self-exploration.

where $t_i^\star$ and $c_i^\star$ are thought and code for the $i$-th step, and the history $h_i = \{t_1, c_1, o_1, \cdots, t_{i-1}, c_{i-1}, o_{i-1}\}$ is composed of thought $\{t_{1,\ldots,i-1}\}$, code $\{c_{1,\ldots,i-1}\}$, and observation $\{o_{1,\ldots,i-1}\}$ of previous steps.

Agent tuning aims to update $\theta$ to increase the tool usage capabilities of agents. This paper proposes an iterative tool usage exploration method, SPORT, to update $\theta$ via step-wise preference optimization in refining trajectories, as shown in Figure 1. Concretely, SPORT has iterative components: task synthesis, step sampling, step verification, and preference tuning. In one iteration, SPORT first generates some multimodal tasks. For each generated task, SPORT performs step sampling and step verification alternately to construct step-wise preference data. Finally, SPORT uses the step-wise preference data to tune the controller.

## 3.2 Task Synthesis

Since a multimodal task is composed of a language query and multimodal files, the task synthesis component is divided into query generation and multimodal file generation. We first generate queries and then generate files, rather than the reverse order, since the multimodal files are diverse (such as DOCX, PPTX, XLSX, and PDF), and it is challenging to construct a diverse file dataset in advance. In addition, tasks are usually based on multiple files instead of only one. First obtaining files and then generating queries may cause weak relevance of files and unnatural queries.

To produce diverse and practical queries, we collect seed from the existing method MAT [18] and employ an LLM (*e.g.*, Qwen2.5-7B) to generate queries. We feed randomly sampled seed queries, used tools, and a designed prompt to the LLM that generates multiple queries at once. For each generated query, we prompt the LLM to output the needed files. If images are needed, we search for source images from off-the-shelf datasets based on similarities. For other files, we prompt the LLM to generate Python code to produce files.

## 3.3 Data Construction

The two components: step sampling and step verification, are performed to construct high-quality preference data, playing key roles in our method. To avoid potential bias issues in constructed data, we introduce an online search scheme, where the step sampling and step verification are executed alternately in each generated task, as shown in Figure 2.

The step-wise preference data is formulated as a triplet $(x_i, a_i^{pre}, a_i^{dis})$, where $x_i = \{Q, F, h_i\}$ denotes the input including the query $Q$, files $F$, and the history $h_i$ of previous steps, $a_i^{pre}$ is the preferred action in the current step, and $a_i^{dis}$ is the dispreferred action.

Concretely, given a task with the query $Q$ and files $F$, the agent expands the search space for the first step by sampling $n$ actions $\{a_1^1, a_1^2, \cdots, a_1^n\}$, including the thought and code $\{t_1^1, c_1^1, \cdots, t_1^n, c_1^n\}$ from the controller, and execute them to obtain $n$ observations $\{o_1^1, \cdots, o_1^n\}$. Then, we feed the query $Q$, $n$ actions, and $n$ observations to an LLM, and ask an LLM to select the best action $\{t_1^*, c_1^*\}$ with its corresponding observation $o_1^*$.

Along $\{t_1^*, c_1^*\}$ and $o_1^*$, we expand the search space for the second step. Regarding $\{t_1^*, c_1^*, o_1^*\}$ as the history $h_2$, the controller samples $n$ actions $\{t_2^1, c_2^1, \cdots, t_2^n, c_2^n\}$ from the controller, and executes them to obtain $n$ observations $\{o_2^1, \cdots, o_2^n\}$. We feed the query $Q$, history $h_2$, $n$ actions in this step, and $n$ corresponding observations to the LLM, and ask the LLM to select the best action $\{t_2^*, c_2^*\}$ with its corresponding observation $o_2^*$. In this case, the agent gradually expands the search space and selects the best action for the next step, until the agent believes that the task is over. Here, we provide observations $\{o_2^1, \cdots, o_2^n\}$ and prompt the verifier to check which tools lead to desirable observations, instead of verifying which observation is correct.

Assume there are $m$ steps in solving one task. In this case, we could collect $m(n-1)$ preference data pairs. In the $i$-th step, the selected best action $\{t_i^*, c_i^*\}$ is the preferred output, and the rest $n-1$ actions are the dispreferred outputs, collected in a set $\mathcal{D}_i^{dis}$, $|\mathcal{D}_i^{dis}| = n-1$. The preference data in one task is denoted as $\{(x_i, a_i^{pre}, a_i^{dis})\}$, where $a_i^{pre} = \{t_i^*, c_i^*\}$, $a_i^{dis} = \{t_i^j, c_i^j\} \in \mathcal{D}_i^{dis}$, and $i \in [1, m]$.

## 3.4 Preference Tuning

**Objective.** In one iteration, we may generate multiple tasks and construct preference data for them. After that, we denote the obtained preference data set as $\mathcal{D} = \{(x_i, a_i^{pre}, a_i^{dis})\}$. We choose the flexible DPO algorithm,

$$\mathcal{L}(\theta) = -\mathbb{E}_{(x_i, a_i^{pre}, a_i^{dis})) \sim \mathcal{D}}[\log \sigma(\beta \log \frac{\pi_\theta(a_i^{pre}|x_i)}{\pi_{ref}(a_i^{pre}|x_i)} - \beta \log \frac{\pi_\theta(a_i^{dis}|x_i)}{\pi_{ref}(a_i^{dis}|x_i)})], \quad (3)$$

where $\pi_\theta$ is the controller to be updated, $\pi_{ref}$ is the controller for reference (the model after SFT in practice), $\beta$ is the weighting parameter that controls the deviation from the reference controller, and $\sigma(\cdot)$ is the logistic function.

**Training Scheme**. The proposed tool usage exploration is performed after an SFT stage for controllers, since the effectiveness of self-exploration requires the controller to have the ability to generate accurate actions. The SFT stage is the same as MAT [18], where 20K trajectories are used to align the agent controller (Qwen2VL-7B in practice) with desirable outputs. In the self-exploration stage, we use preference tuning to update Qwen2-VL. The preference tuning process is summarized in Algorithm 1.

---

**Algorithm 1:** Training process in SPORT

---

**Require:** Seed of tasks, initial agent controller $\pi_\theta$ after SFT, and $\pi_{ref} = \pi_\theta$. Preference data $\mathcal{D} = \emptyset$.
**Ensure:** Updated agent controller $\pi_{\theta^*}$.
1: **while** Not converged **do**
2:     Set $\mathcal{D} = \emptyset$.
3:     Randomly sample task seeds, and send them to an LLM to generate tasks.
4:     **for** Each generated task **do**
5:         **for** the $i$-step in solving the task **do**
6:             Sample $n$ actions $\{t_i^1, c_i^1, \ldots, t_i^n, c_i^n\}$ based on the history $h_i$, and execute them to obtain results $\{o_i^1, \ldots, o_i^n\}$.
7:             Select the best action $\{t_i^\star, c_i^\star\}$.
8:             Construct $n-1$ preference pairs, and add them into $\mathcal{D}$.
9:             Add $t_i^\star, c_i^\star, o_i^\star$ into $h_i$.
10:        **end for**
11:     **end for**
12:     Use $\mathcal{D}$ to update $\pi_\theta$ via the preference tuning algorithm in Eq. (3).
13: **end while**

---

Table 1: Results on two benchmarks: GTA and GAIA. The **bold** results represent the best performance compared to the open-source models.

| Method | Controller | GTA | | | GAIA | | | |
|--------|-----------|-----|-----|-----|------|-----|-----|-----|
| | | *ToolAcc* | *CodeExec* | *AnsAcc* | *Level 1* | *Level 2* | *Level 3* | *AnsAcc* |
| *Closed-source Controller* | | | | | | | | |
| Lego Agent | GPT-4 | - | - | 46.59 | - | - | - | - |
| Lego Agent | GPT-4o | - | - | 41.52 | - | - | - | - |
| Warm-up Agent | GPT-4-turbo | - | - | - | 30.20 | 15.10 | 0.00 | 17.60 |
| HF Agent | GPT-4o | 63.41 | 95.12 | 57.05 | 47.17 | 31.40 | 11.54 | 33.40 |
| HF Agent | GPT-4o-mini | 56.10 | 100.00 | 57.69 | 33.96 | 27.91 | 3.84 | 26.06 |
| *Open-Source Controller* | | | | | | | | |
| HF Agent | LLaVA-NeXT-8B | 14.97 | 25.08 | 14.10 | 9.43 | 1.16 | 0.00 | 3.64 |
| HF Agent | InternVL2-8B | 36.75 | 52.18 | 32.05 | 7.55 | 4.65 | 0.00 | 4.85 |
| HF Agent | MiniCPM-V-8.5B | 36.59 | 56.10 | 33.97 | 13.21 | 5.81 | 0.00 | 7.27 |
| HF Agent | Qwen2-VL-7B | 44.85 | 65.19 | 42.31 | 16.98 | 8.14 | 0.00 | 9.70 |
| T3-Agent | MAT-MiniCPM-V-8.5B | 65.85 | 80.49 | 52.56 | 26.42 | 11.63 | **3.84** | 15.15 |
| T3-Agent | MAT-Qwen2-VL-7B | 64.63 | 84.32 | 53.85 | 26.42 | 15.12 | **3.84** | 16.97 |
| *Ours* | | | | | | | | |
| SPORT Agent | Tuned-Qwen2-VL-7B | **72.41** | **91.87** | **60.26** | **35.85** | **16.28** | **3.84** | **20.61** |

## 4 Experiments

### 4.1 Setting

The performance of the proposed SPORT approach is assessed on the GTA [51] and GAIA [41] benchmarks. Results are compared against agents powered by both closed-source models (*e.g.*, GPT-4, GPT-4o, Claude3) and open-source models, including LLaMA-3-70B-instruct [12], Qwen1.5-72B-chat [3], LLaVA-NeXT-8B [33], InternVL2-8B [8], Qwen2-VL-7B [52], and MiniCPM-V-8.5B [58]. Specifically, we perform direct comparisons with leading agents, such as Lego Agent [1] and Warm-up Act Agent [41]. As a baseline, we use the Huggingface Agent (HF Agent) [22], which operates with the same toolset as the SPORT Agent . We first evaluate these agents on two benchmarks, then assess the quality of the produced preference data, and finally show several visualization examples to demonstrate the effectiveness of our method.

We employ the Qwen-2-VL model as the controller. In the training process of our VLM controller, we freeze the vision encoder and visual token compressor, and fine-tune the language model using LoRA [20]. We set the rank as 32 and apply LoRA on query, key, and value projection matrices in all self-attention layers. We use the AdamW optimizer with a cosine annealing scheduler. The learning rate is $1.0e - 6$ and the batch size is 2 per device. We set the max context window as 10240 to support complex trajectories of our agent. All training is conducted on a node equipped with $8 \times$ A100 GPUs. The training time is positively correlated with the number of iterations and iteration step size $d$. For all the evaluations, we *disable* the sampling and verification during inference for fair comparison.

**Benchmark.** The GTA and GAIA benchmarks serve as robust evaluation frameworks for assessing multimodal agents. The GTA benchmark includes 229 tasks paired with 252 images, where task completion requires 2 to 8 steps, with most tasks involving 2 to 4 steps. This benchmark challenges multimodal agents to exhibit advanced perception, operational skills, logical reasoning, and creative thinking based on visual data. In real-world multimodal scenarios, agents often need to handle diverse file formats such as PPTX, PDF, and XLSX. To evaluate agent performance on such files, the GAIA benchmark is used, comprising 446 tasks across 109 files. GAIA's tasks are organized into three levels, with task complexity varying from 2 steps to sequences of indefinite length. It evaluates document comprehension, web navigation, logical reasoning, and summarization abilities.

**Metric.** Following existing methods [51, 18], we assess agent performance using three key metrics: *AnsAcc*, *ToolAcc*, and *CodeExec* for the GTA benchmark. *AnsAcc* gauges the accuracy of predicted answers. *ToolAcc* evaluates the correctness of tool usage and the quality of answer summaries. *CodeExec* measures the percentage of generated code that executes without errors. In the GAIA benchmark, we focus on measuring *AnsAcc* at its three levels.

### 4.2 GTA Results

The results on the GTA benchmark are shown in Table 1, where key metrics including *AnsAcc*, *ToolAcc*, and *CodeExec* are reported. Our agent surpasses the Lego agent that utilizes closed-source

models (*e.g.*, GPT-4 and GPT-4o), as well as the HF agent that uses closed-source models and open-source models (*e.g.*, InternVL2-8B), showcasing the ability of our SPORT Agent to tackle complex tasks with greater efficiency. A comparison between agents through SFT (*i.e.*, T3-Agent) and our SPORT Agent demonstrates the effectiveness of our online self-exploration framework and the advantages of our Step-wise optimization approach. Our method has about 7% improvements on the final accuracy, since it calls more suitable tools (8% improvements) and reduces code error (7% improvements). Compared with the HF agent using GPT-4o and GPT-4o mini, our agent achieves higher *ToolAcc* and comparable *CodeExec*. This indicates that the proposed SPORT method improves the planning and reasoning capabilities of agents again.

### 4.3 GAIA Results

In Table 1, we report the performance of SPORT Agent on the GAIA validation set. SPORT Agent achieves best results among agents that use open-source models, surpassing the best-performing open-source model, Qwen2-VL-7B, by about 11% on *AnsAcc*. The consistent improvements across different levels underscore the efficacy of our online self-exploration framework. Furthermore, SPORT Agent demonstrates significant gains over the SFT-tuned controller, with an improvement of about 4% over MAT-Qwen2-VL-7B. However, when compared to agents leveraging closed-source models such as GPT-4, SPORT Agent exhibits a slight performance gap. We attribute this discrepancy to the larger model sizes and more extensive training data available to closed-source models.

### 4.4 Ablation

**Effectiveness of Iteration Step Size**   We conduct an ablation study on the GTA benchmark to investigate the impact of the iteration step size $d$ that denotes the number of used trajectories in each iteration. We set $d \in \{200, 500, 1000\}$, adjusting the number of iterations to (5, 2, 1), respectively, to ensure a total of 1000 trajectories are processed in each setting. As shown in Table 2, setting $d = 500$ yields the best overall performance across all metrics. When $d = 1000$, the model sees all tasks in a single pass, which limits adaptability and leads to a slight drop in answer accuracy and execution success. In contrast, using a smaller step size ($d = 200$) results in less diverse updates per iteration, which may reduce robustness. These results suggest that a moderate value of $d$ offers a good balance between update frequency and data diversity, leading to more stable and effective training.

**Comparison with Static Preference Data**   We conduct ablation experiments to compare our method (online exploration) with DPO on static tasks. In doing so, we directly apply DPO to the MM-Traj dataset [18]. Specifically, we treat the MM-Traj data (GPT-4o generated) as "preferred" samples and synthetically generate an equal number of "dispreferred" examples via the MAT-SFT model, matching the total volume of our SPORT preference data. We then fine-tune MAT-SFT using DPO under this constructed preference dataset.

Table 3 reports results on the GTA benchmark. Compared to vanilla MAT-SFT, MAT-SFT-DPO yields only modest improvements (AnsAcc +1.28, ToolAcc +2.67, CodeExec +1.58), indicating that naïvely applying DPO to MAT provides limited gains. In contrast, SPORT substantially outperforms MAT-SFT-DPO (AnsAcc +5.13, ToolAcc +5.11, CodeExec +5.97), demonstrating the effectiveness of our framework in leveraging diverse, multimodal preference data.

**Training with Different Base Models**   To evaluate the generalizability of SPORT, we apply it to different base models. We compare four configurations on the GTA benchmark: Qwen2-VL-7B (base model), MAT-Qwen2-VL-7B (base model with MAT applied), SPORT-Qwen2-VL-7B (SPORT applied directly to base model), and SPORT-MAT-Qwen2-VL-7B (SPORT applied to MAT-tuned model).

Table 2: Ablation on iteration step $d$ in the GTA benchmark.

| $d$ | AnsAcc (%) | ToolAcc (%) | CodeExec (%) |
|-----|-----------|-------------|--------------|
| 200  | 56.41 | 68.58 | 88.46 |
| 500  | 57.69 | 69.87 | 89.74 |
| 1000 | 57.05 | 69.87 | 88.46 |

Table 3: Ablation on preference data: MAT-SFT *vs.* MAT-SFT-DPO *vs.* SPORT on the GTA benchmark.

| Method | AnsAcc (%) | ToolAcc (%) | CodeExec (%) |
|--------|-----------|-------------|--------------|
| MAT-SFT     | 53.85 | 64.63 | 84.32 |
| MAT-SFT-DPO | 55.13 | 67.30 | 85.90 |
| SPORT (Ours) | **60.26** | **72.41** | **91.87** |

Table 4: Answer accuracies (%) on GTA benchmark with different base models.

| Model | AnsAcc (%) |
|---|---|
| Qwen2-VL-7B | 42.31 |
| MAT-Qwen2-VL-7B | 53.85 |
| SPORT-Qwen2-VL-7B | 55.13 |
| SPORT-MAT-Qwen2-VL-7B | 60.26 |

Table 5: Impact of task diversity on the GTA benchmark.

| Method | AnsAcc (%) |
|---|---|
| MAT-Qwen2-VL-7B | 53.85 |
| SPORT w/ 5 from 100 seeds | 58.33 |
| SPORT w/ 20 from 425 seeds | 60.26 |

Table 4 presents the answer accuracies. Applying SPORT directly to Qwen2-VL-7B improves accuracy from 42.31% to 55.13% (+12.82%), demonstrating SPORT's effectiveness as a standalone method. When applied to MAT-tuned baseline, SPORT achieves the highest accuracy of 60.26% (+5.13% over MAT alone), indicating that SPORT can effectively complement existing tuning methods. These results confirm that SPORT is a flexible approach that enhances agent performance both independently and in combination with other preference optimization methods.

**Sensitivity to Task Quality and Diversity**  To assess SPORT's robustness to the quality and diversity of synthetic tasks in early-stage self-exploration, we conduct an ablation study where the in-context examples are reduced from 20 to 5 and the task seed pool is narrowed from 425 to 100. This setup mimics a low-diversity scenario during early-stage exploration.

As shown in Table 5, this reduction leads to a moderate drop in performance (60.26 → 58.33). However, the result still significantly outperforms the SFT baseline (58.33 vs. 53.85), demonstrating that SPORT maintains strong performance even under constrained task diversity. These findings indicate that while task quality and diversity do impact SPORT's effectiveness, the method exhibits robustness to variations in the synthetic task generation process.

## 4.5 Statistic

We aggregate step-wise preference data from all iterations, each with $d = 500$ and a sampling size of 5, resulting in a total of 16K samples. We analyze the differences between the *chosen* and *rejected* step-wise data in terms of code error rate, tool selection, and content variations.

**Tool Distribution**  We analyze the distribution of tools in the chosen and rejected steps by examining their frequency of occurrence, as shown in Figure 3. The results highlight differences in tool usage between the two groups.

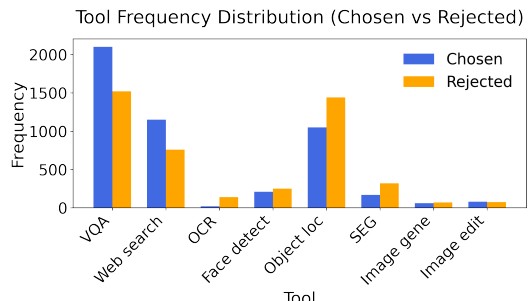

In the chosen steps, tools such as `visualizer` (2101 occurrences) and `objectloc` (1051 occurrences) are used more frequently. In contrast, in the rejected steps, `objectloc` (1442 occurrences) and `visualizer` (1524 occurrences) are more prevalent. Additionally, tools like `ocr` and `seg` appear more often in the rejected steps.

Figure 3: Tool distribution for the chosen and rejected steps.

To quantify the overall discrepancy in tool usage, we computed a tool distribution difference of $45.62\%$, indicating a substantial variation in the tool selection between the chosen and rejected steps.

**Code Error Rate**  We compared the code execution status of chosen and rejected steps by measuring the proportion of execution results (Observations) that contained code errors. The rejected steps exhibit a significantly higher error rate (81.94%) compared to the chosen steps (18.35%). This indicates that our step-wise preference data favors code that executes successfully. Consequently, this preference also leads to the improvement in the code accuracy of the SPORT Agent, as shown in Table 1 ($CodeExec$ 84.32% → 91.87%).

**Content Difference**  We compared the BLEU scores of steps selected by our verifier and those selected randomly. BLEU scores measure the similarity between different sequences, with lower scores indicating higher discrimination. As shown in Table 6, our verifier consistently achieved

Table 6: Comparison of BLEU scores (lower scores indicate greater discrimination) between our verifier and random selection.

| Verifier | B1 ($\downarrow$) | B2 ($\downarrow$) | B3 ($\downarrow$) | B4 ($\downarrow$) |
|---|---|---|---|---|
| Random Select | 0.53 | 0.41 | 0.36 | 0.34 |
| Ours | 0.30 | 0.18 | 0.14 | 0.11 |

Table 7: Average scores from humans on data quality.

| Task | | Trajectory | | | Preference |
|---|---|---|---|---|---|
| Reasonable | Natural | Tool | Content | Code | |
| 8.16 | 8.48 | 8.78 | 9.08 | 8.44 | 82% |

lower BLEU scores across all n-grams compared to random selection. Specifically, for BLEU-1, BLEU-2, BLEU-3, and BLEU-4, our verifier's scores were 0.30, 0.18, 0.14, and 0.11, respectively, while random selection yielded higher scores of 0.53, 0.41, 0.36, and 0.34. The results demonstrate that our verifier selects more distinct steps, enhancing the quality of the chosen steps.

## 4.6 Data Quality

To evaluate the effectiveness of the constructed preference data, we conducted a user study involving 20 AI researchers with coding and development experience from various universities and research institutes. These participants were not provided with any background information about our methodology; instead, they were only briefed on the purpose and functionality of the agent. They were required to justify whether the preferred and dispreferred pairs were proper.

The evaluation was performed for tasks and trajectories, based on five criteria: (1) 'Reasonableness' to evaluate whether the generated tasks are infeasible; (2) 'Naturalness' to evaluate whether the generated tasks are natural; (3) 'Code' to evaluate the accuracy of code in action; (4) 'Tool' to assess the appropriateness of tool selection; and (5) 'Parameter' to assess the correctness of parameter passing. Participants provided scores ranging from 1 to 10, with higher scores indicating better performance. As shown in Table 7, the average scores for our task and framework exceeded 8, demonstrating the validity of the preference data collected by our approach.

To further assess the effectiveness of the verifier in generating preference data, we conducted an additional study with a separate group of 20 researchers. Each participant was asked to evaluate 50 steps sampled in parallel by making preference selections. The preferences were determined using the same three criteria: code accuracy, tool selection appropriateness, and parameter passing correctness. We measured the agreements between the verifier's preferences and those of the human participants, denoted as 'Preference' in Table 7. The results revealed an 82% overlap. This high level of agreement validates the reliability of the verifier in capturing human-like preferences.

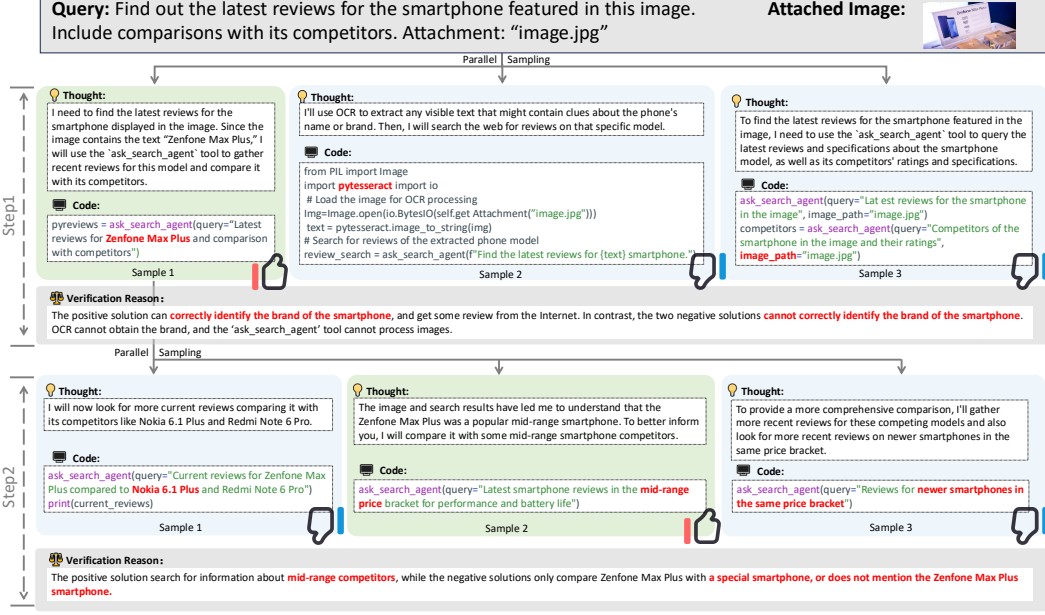

Figure 4: Example of constructed preference data.

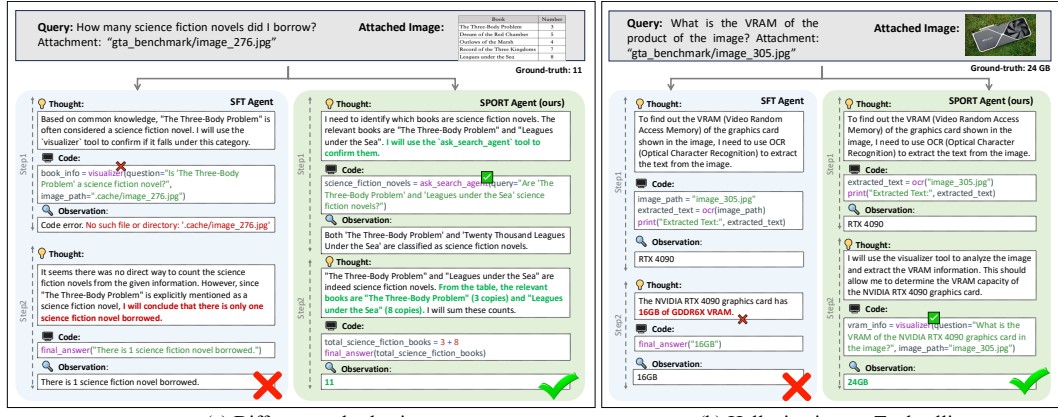

(a) Different tool selection.   (b) Hallucination *vs*. Tool calling.

Figure 5: Comparisons between SFT Agent and our SPORT Agent.

## 4.7 Visualization

We visualize the preference data generated by the online self-exploration framework, as shown in Figure 4. The verifier can successfully choose actions that lead to correct intermediate results. In step 1, the action that produces the correct brand and basic information about the smartphone is selected as the preferred data. In step 2, the action that searches for content most relevant to the task is selected as the preferred data. It compares the smartphone with mid-range competitors, while the rest compares the smartphone with a special one.

We visualized the task-solving procedure of the SPORT Agent compared with the SFT Agent (T3 agent), as shown in Figure 5. The SPORT Agent after the step-wise preference tuning can well solve the issues of code hallucination and tool error. For example, in case (a), the SPORT Agent correctly selects the web search tool while the SFT Agent uses the wrong tool with an incorrect image path. In case (b), the SPORT Agent utilizes tools to solve the task, while the SFT Agent produces an answer via hallucination.

## 5 Conclusion

In this paper, we have presented an online self-exploration framework for multimodal agents, through which the agents can learn via automatic interaction with new environments without accessing any annotations. Based on this framework, we have presented a step-wise optimization for refining trajectories (SPORT), which can produce in-distribution preference data in complex environments. Given proper prompts, the proposed SPORT method can generate diverse multimodal tasks and provide good verification of agent actions aligned with humans. Experiments on two challenging benchmarks, GTA and GAIA, show that the proposed SPORT method achieves significant improvements on multimodal agents, demonstrating its effectiveness.

**Limitations** The verifier plays an important role in the current SPORT method. However, it heavily relies on human-designed rules and prompts, causing inferior generalization for some outliers. In the future, we will explore the self-exploration techniques for the verifier, that is, learning to verify, through which the verifier can adapt to new environments with the controller together. Furthermore, we will explore the theoretical guarantee for the verifier, allowing it to scale to open settings.

**Acknowledgements** This work was supported by the Natural Science Foundation of China (NSFC) under Grants No. 62406009, No. 62172041 and No. 62176021, Shenzhen Science and Technology Program under Grant No. JCYJ20241202130548062, and Natural Science Foundation of Shenzhen under Grant No. JCYJ20230807142703006.

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

# A  Broader Impacts

SPORT's ability to autonomously explore tool usage and refine behavior via preference feedback promises to lower the barrier to creating versatile multimodal agents, enabling researchers and practitioners—even those with limited resources—to build domain-adapted systems for tasks such as document analysis, scientific data interpretation, and accessible educational or healthcare interfaces. By reducing reliance on costly human annotation and manual rule-crafting, SPORT can accelerate innovation across diverse fields, fostering more inclusive and scalable AI solutions.

At the same time, increased agent autonomy raises risks of unintended bias, error propagation, and resource inefficiency. SPORT's verifier, grounded in heuristic rules and model judgments, may inadvertently reinforce spurious behaviors or hallucinations, particularly in high-stakes domains like legal or medical analysis. We therefore advocate for transparent auditing, human-in-the-loop oversight, and careful management of exploration budgets to ensure responsible deployment.

# B  Comparison with Existing Sampling Frameworks

Our method constructs step-level preference data for complex multimodal tasks. To contextualize its contributions, we compare against representative reinforcement-learning–based sampling frameworks along three dimensions: (1) **Task Domain**, (2) **Collection Granularity**, and (3) **Annotation Format**. Table 8 summarizes this comparison.

Table 8: Comparison with existing sampling schemes

| Method | Task Domain | Collection Granularity | Annotation Format |
|---|---|---|---|
| WebRL [44] | GUI control | Trajectory-Level | Finetune a reward model |
| PAE [65] | GUI control | Trajectory-Level | Use a pre-trained model |
| ETO [47] | GUI control & Embodied AI | Trajectory-Level | Expert labels for comparisons |
| DMPO [46] | GUI control & Embodied AI | Trajectory-Level | Finetune a reward model |
| DigiRL [2] | GUI control | Step-Level | Finetune a reward model |
| TP-LLAMA [7] | API calling | Step-Level | Use expert data |
| IPR [56] | GUI control & Embodied AI | Step-Level | Use expert data |
| StepAgent-Inverse [11] | GUI control & Embodied AI | Step-Level | Use expert data |
| **Ours** | **Multimodal Reasoning** | **Step-Level** | **Use a pre-trained model** |

The multimodal reasoning domain poses unique challenges that prior sampling frameworks do not adequately address:

1. **Data scarcity.** There is a shortage of collected tasks and expert trajectories in complex multimodal settings.

2. **Inadequate reward modeling.** Approaches effective in GUI or API domains—such as pre-trained classifiers or rule-based rewards—struggle to capture multimodal task complexity.

3. **Low sampling efficiency.** Generating high-quality trajectories for multimodal tasks requires expensive tool usage (e.g. LLM calls, web searches), making naive sampling impractical.

To overcome these issues, we leverage pre-trained LLMs to generate step-wise preference data automatically, providing a scalable and practical solution for training agents on complex multimodal reasoning tasks.

# C  Computational Efficiency Analysis

We provide a comprehensive comparison of compute time and GPU usage between the baseline method (MAT) and our SPORT framework. Table 9 summarizes the estimated compute time and GPU cost for both methods, measured in GPU hours (GPUh) using a single A100 80GB GPU.

**Data Generation Efficiency.** The computational cost of Task Synthesis, Step Sampling, and Tool Calling is closely tied to the data scale. MAT synthesizes and samples steps for 20K tasks and 20K trajectories, resulting in high overhead across all three components. In contrast, SPORT performs Task Synthesis on only 2K tasks and constructs 16K preference pairs for tuning, leading to substantially reduced compute time in these stages. Specifically, SPORT reduces Task Synthesis time from 29.13h to 3.61h, Step Sampling from 69.92h to 4.53h, and Tool Calling from 58.93h to 9.86h.

Table 9: Estimated compute time and GPU cost comparison between MAT and SPORT. GPU hours (GPUh) are measured as usage of a single A100 80GB GPU for one hour. The estimates for MAT are derived from its reported tool invocation frequency, combined with empirical measurements of the GPT-4o-mini API and tool execution latency on our hardware.

| Component | MAT Time (h) | SPORT Time (h) | MAT Cost (GPUh) | SPORT Cost (GPUh) |
|---|---|---|---|---|
| *Data Generation* | | | | |
| Task Synthesis | 29.13 | 3.61 | 0 (GPT-4o-mini API ~$500) | 3.61 (Qwen2-VL-7B) |
| Step Sampling | 69.92 | 4.53 | 0 (GPT-4o-mini API ~$1500) | 4.53 (Tuned-Qwen2-VL-7B) |
| Step Verification | 0 | 3.36 | 0 | 3.36 (Qwen2.5-7B) |
| Tool Calling | 58.93 | 9.86 | 58.93 | 9.86 |
| *Training* | | | | |
| Model Training w/ 4× A100 80G GPU | 15.77 | 15.20 | 63.08 | 60.80 |
| **Total** | **173.75** | **36.56** | **122.01** | **82.16** |

**Step Verification.** SPORT introduces an additional Step Verification process to obtain step-level preferences, which incurs 3.36 GPU hours. While this stage adds computational cost, it remains acceptable relative to the total runtime due to the smaller data volume (16K preference pairs vs. 20K trajectories in MAT).

**Training Efficiency.** Model training with 4× A100 80GB GPUs shows comparable efficiency between both methods, with SPORT requiring 15.20h (60.80 GPUh) compared to MAT's 15.77h (63.08 GPUh). This marginal difference demonstrates that SPORT's efficiency gains primarily stem from the data generation stage rather than the training phase.

**Overall Cost Reduction.** As shown in Table 9, SPORT achieves a total compute time of 36.56h compared to MAT's 173.75h, representing a **4.75× speedup**. In terms of GPU cost, SPORT requires 82.16 GPUh versus MAT's 122.01 GPUh, resulting in a **32.7% cost reduction**. These substantial improvements in computational efficiency, combined with SPORT's superior performance, validate the effectiveness of our preference-based learning approach.

# D  Task Generation

Following MAT [18], we first use an LLM to generate *queries*, and then generate both the *file content* and *file type* based on the query. Depending on the *file type*, we adopt different strategies for file generation:

- For *image files*, we retrieve relevant images from a large image dataset [6] based on the generated file content.

- For *non-image files*, `.PDF`, `.XLSX`, `.DOCX`, or `.MP3`, etc, we employ the LLM to write Python code that calls relevant libraries to convert the file content into the corresponding file format.

After generating the files, we adopt a two-step query-file verification process—*Revision* and *Filtering*—to ensure the quality of each task (i.e., query-file pair).

In the *Revision* step, both the query and the corresponding file are fed into a vision-language model (VLM), which is instructed to revise the query to better align with the file if necessary. For image data, we directly input the image into the VLM; for non-image data, we input the file content instead.

In the *Filtering* step, the VLM is no longer allowed to modify the query. Instead, it is asked to assess whether the task meets a predefined quality threshold. Only tasks that pass this quality check are retained, while all others are discarded.

## D.1  Model Comparison on Task Generation

We conducted a comparative analysis of task quality between open-source and closed-source models. For this evaluation, we employed Qwen-2VL-7B (open-source) and GPT-4o-mini (closed-source) to generate 200 tasks each under identical system prompts. The resulting 400 tasks were subsequently randomized and anonymized to eliminate source bias. We recruited 20 human evaluators, with each participant assessing 20 tasks according to two key metrics: naturalness and reasonableness, rated on a 10-point scale (higher scores indicating superior quality). As demonstrated in Table 10, the tasks

produced by both models achieved comparable quality ratings, providing compelling evidence that open-source models possess sufficient capability for high-quality task generation in this domain.

Table 10: User study for open-source *vs.* close-source models generated tasks. Scores are scaled from 1 to 10 and a higher score denotes better quality.

| Model | Task Naturalness | Task Reasonableness |
|---|---|---|
| GPT-4o-mini | 8.71 | 8.37 |
| QWen-2VL-7B | 8.75 | 8.35 |

# E    Error bar for the main results

We conduct each experiment five times and report the performance variance, as shown in Table 11. The results demonstrate that our improvements are statistically significant compared to the observed variances.

Table 11: Performance with variance on the GTA benchmark.

| Method | Controller | *AnsAcc* | *ToolAcc* | *CodeExec* |
|---|---|---|---|---|
| T3-Agent | MAT Tuned Qwen2-VL-7B | 53.85 | 64.63 | 84.32 |
| SPORT Agent (Ours) | Tuned Qwen2-VL-7B | 60.26 ± 1.51 | 72.41 ± 1.11 | 91.87 ± 1.41 |

# F    System Prompts

We referenced the task generation prompts from MAT[18] for our implementation.

## F.1    Prompt for Query Generation

The prompt for query generation is shown in fig. 6.

> You are tasked with generating user queries that will prompt an agent to call various tools (only use the tool listed in our toolset), including internet search capabilities, to solve real-world, practical problems. The problems should be natural, varied, and challenging, requiring the agent to reason across different domains. Ensure that the problems span a range of practical scenarios.
>
> Our toolset: TOOL_SET
> I will now provide examples, along with the tools.
> Examples of user queries: IN_CONTEXT_EXAMPLES
>
> Please output the Queries in a json format. Make sure that the queries share a similar style to the in-context examples. The output template is:
> {
>  "query": "What is the weather today?", <The user query to the agent.>
>  "tools": ["tool1", "tool2",...], <A list consisting of the tool names related to the query.>
> },
> ...

Figure 6: Prompt for query generation.

## F.2    Prompt for File Generation

The prompt for file content generation is shown in fig. 7 and fig. 8, and the prompt for file code generation is shown in  fig. 9 and  fig. 10.

You are a smart reasoner that can restore a query_solving scene between a human and an agent. Human gives a complex query and several images to the agent, and then the agent answers the query by searching on the Internet and applying tools to the images with step-by-step reasoning. Now, you will be given the query with suggested tools, I suggest you analyze the needed information to solve the query, and divide the information into two groups: searching from the Internet and extracting from the images using tools. Based on the information from the images, you need to further infer the content of these images, through which the agent could correctly solve the query.

Our toolset: TOOL_SET
Output MUST use the following json template.

{
    "information": <Needed information to solve the query. For the query including creating/generating images, the information should NOT be the description of the described image.>
    "information from the Internet": <Information from the Internet inferences based on the given query and suggested tools. Determine which information is suitable to be obtained from the Internet. Or say no information is required from the Internet.>
    "information from images": <Information extracted from given images based on the suggested tools to solve the query. It should be several sentences, including information extracted from the images using tools. Determine which information is suitable to be obtained from the images, and using which tools. Do not generate image_content for the query including generating/creating an image. Or say no information is required from the images.>
    "file": {
        "image_numbers": <set an int number, the number is depended on needed information from images>,
        "image_content":
        {
            "image_1": <The image content should be a natural language, describing the content of the
first
            first image relevant to the query. The content should be concrete, such as concrete
            number, concrete name. The content should match the query and the above images.>
            ... <if you think the query needs more than 1 image, please output image content like
            'image_2'.>
        }
    }
}

Figure 7: System prompt for the file content generation.

Now given the query: QUERY, firstly analyze the needed information to solve the query and divide the information into two groups: searching from the Internet or extracting from images using tools. Then for information from images, imagine possible answers for each information (it should be concrete answers instead of descriptions). Finally, output the json for the inferenced information and the content of images.

Figure 8: User prompt for the file content generation.

You are a helpful assistant and can generate a <file type placeholder> file by writing Python code. You will be given a description of the content of the file. You need to first largely extend the content, and then write Python code to generate a <file type placeholder> file. GUARANTEE that the provided content is in the file.
The output Python code MUST use the following template.
"""
        ## extention start
                Extened content: <here is the extented content>
        ## extention end

        ## code start
            <here is the Python code to generate a <file type placeholder> file>
        ## code end
"""

Figure 9: User prompt for the *non-image* file generation.

Now, given the following content: <file content>, first largely extend the content, and output a code to generate a <file type placeholder> file, where the file name is <file name> and the file will be saved in <save path>.

Figure 10: User prompt for the *non-image* file content generation.

### F.3   Prompt for Query-file Filter

The prompt for the query-file filter is shown in fig. 11 and fig. 12.

### F.4   Prompt for Step Verifier

To evaluate the quality of intermediate steps taken by an agent during task execution, we design a step verifier consisting of two key components: a system prompt and a user prompt. The system prompt (Figure 13) provides the verifier model with detailed instructions for evaluating multiple candidate steps ('CURRENT_STEP') based on their coherence, logical progression, and effectiveness in advancing the task. It guides the model to consider contextual alignment with the prior step, tool usage, hallucination, and content relevance. The verifier is required to select the best step and justify its decision in a structured json format.

The user prompt (Figure 14) supplies the concrete input to the verifier, including the task description and a list of candidate step sets. Each step set contains the result from the previous step, the current step (thought and code), and the result produced by executing that step. Together, these prompts simulate a human-like evaluation process, encouraging the model to perform judgment aligned with human preferences in multi-step reasoning scenarios.

You are a helpful assistant that is given a query and several images. You need to check whether the images are relevant to the query. The query and images are used to evaluate the perception ability, reasoning ability, and information search ability of an AI agent. The agent solves the query by searching for information on the Web and extracting information from the images. In some cases, based on the given images, the agent could not solve the query, even though it searched for information from the Web (e.g., some specific knowledge). You need to pick up these bad cases.

The agent can call the following tools to solve the query. TOOL_SET .

Thus, the images should follow these requirements.
1. Relevance: The depicted scenarios or objects in images should be relevant to the query. The images should contain scenarios or objects that are mentioned in the images.
2. Usefulness: The image should contain necessary information to address the query, such as some specific details that cannot be obtained from the Web.
3. Some queries require the agent to search for knowledge from the Web and combine the information in the image to solve the queries. Thus, in some cases, the images do not contain all the information to solve the query, but the missed information could be searched on the Web. These cases should be regarded as correct cases.

The output MUST use the following json template to check the images.
{     "information_for_query": <Required information to solve the query.>,
     "useful_information_in_image": <Useful information that can be extracted from images to solve the query>,
     "missed_information_in_images": <Missed information that is necessary to solve the query but does not exist in the images.>,
     "missed_information_web_search": <You need to justify whether the missed information could be searched from the Web, using your rich experience in surfing the Internet.> ,
     "missed_information_obtained": <You need to justify whether the missed information could be obtained via computing or reasoning based on information extracted from the images or searched from the Web.>,
     "thought": <Now, you need to determine whether the images can solve the query. If the missed information could be searched from the Web or obtained based on existing information, the images can solve the query. If not, the images cannot solve the query.>,
     "correct": <According to the above reasoning, if you consider the images reasonable for the query to be solved by the tools, set the value to 'yes', otherwise set the value to 'no'.>,
     "updated_query": <If you judge the correctness as 'no', please rewrite the query to make it more relevant to the given images. If you judge the correctness as 'yes', please output "no revision is needed." >
} '''

Figure 11: System prompt for the query-file verification.

Following are images, the query: <query>, inference whether the images can solve the query based on the perception ability, reasoning ability, and information search ability of an AI agent.

Figure 12: User prompt for the query-file verification.

You are an evaluation assistant responsible for analyzing and evaluating agent trajectories. Your goal is to rank <N> 'CURRENT_STEP' entries based on their coherence, logical progression, and effectiveness in addressing the TASK, as observed in the 'CURRENT_RESULT', and their alignment with the 'PREVIOUS_STEP'.

Input Description:
You will receive <N> sets of the following:
    - 'PREVIOUS_RESULT': The prior results obtained by the agent.
    - 'CURRENT_STEP': The agent's output, containing a 'thought' and 'code' intended to complete the task based on the observation.
    - 'CURRENT_RESULT': The result or state produced by executing the 'CURRENT_STEP'.

Your Task:
1. Evaluate each 'CURRENT_STEP':
    - Assess how well the proposed 'CURRENT_STEP' aligns with the context established by the 'PREVIOUS_STEP' and the observation reflected in the 'CURRENT_RESULT'.
    - Check for coherence, logical progression, and contextual relevance.
    - Prioritize outputs that effectively build upon or adapt to the 'PREVIOUS_STEP' while addressing the 'CURRENT_RESULT'.

2. Select the BEST of the 'CURRENT_STEP' entries:
    - Pick the best 'CURRENT_STEP' according to the following guidelines.

3. Provide a concise explanation for your choice:
    - Highlight key factors that influenced your decision, such as logical flow, contextual relevance, effectiveness, and uniqueness of the result.

Evaluation Guidelines:
    - Hallucination: Penalize the directly hallucinated content in the code instead of being produced from tools.
    - Tool selection: Pay attention to whether the controller selects the proper tool.
    - Best content pass into the tool: For the two step that uses the same tool, pay attention to the query that the controller sends to the tools, such as the 'question' in visualizer() and ask_search_agent().
    - Task Relevance: Ensure the CURRENT_STEP contributes meaningfully to solving the task.
    - Maintain objectivity and avoid assumptions beyond the provided inputs.

Output Format:
Return your evaluation in the following json structure:
{
    "reason": "<concise_explanation_of_ranking>"
    "best_id": <An int that indicates the id for the best step. Since there are five CURRENT_RESULTs, the id should only be one of 1,2,3,4, and 5>,
}

Figure 13: System prompt for the step verifier.

The following are the given task, results of previous steps, and result of the current step.

TASK: <task>

Step Sets: <step_set>
# Format of step_set:
# {
#      'PREVIOUS_RESULT': <The prior results obtained by the agent.>
#      'CURRENT_STEP': <The agent's output, containing a 'thought' and 'code' intended to complete the task based on the observation.>
#      'CURRENT_RESULT': <The result or state produced by executing the 'CURRENT_STEP'.>
# }
Now, you need to determine the best of the current steps based on the above information.

Figure 14: User prompt for the step verifier.

# G    User Study Interface

## G.1    Preference Alignment Study

Figure 15a presents the web interface used to evaluate how well our automated verifier's preferences align with those of human judges. In each trial, participants were shown a single task case along with a collection of candidate next-step actions (each consisting of a brief "Thought" description and an optional code snippet or tool invocation). These options were the same ones ranked by our verifier, but presented in random order to prevent positional bias.

Participants were instructed to review each candidate step and select the one they considered most appropriate for progressing the task. No additional scoring rubric was provided: judges were simply asked to choose the option they "would use" if they were guiding the model. Once a selection was made, participants clicked "Submit" to lock in their preference and proceed to the next case.

By comparing the human-selected option against the top choice of the verifier, we compute an *agreement rate* for each model and task type. High agreement indicates that the verifier captures human judgments of step quality; lower agreement reveals areas where the verifier's ranking diverges from human intuition. These results were aggregated over 50 cases per participant, enabling both per-case analysis and overall statistics on human–verifier alignment.

## G.2    Data quality

Figure 15b illustrates the web interface employed in our user study. For each case, participants proceeded through two consecutive scoring phases:

**Task Evaluation.**

- *Reasonableness* (1–10): Does the prompt and the displayed interaction trajectory form a logical, feasible, and well-defined user request?
    - 1–3: Highly unreasonable or ill-posed.
    - 4–6: Somewhat reasonable but with noticeable flaws.
    - 7–9: Mostly reasonable, with only minor issues.
    - 10: Fully logical and indistinguishable from genuine user queries.
- *Naturalness* (1–10): Is the phrasing realistic, user-like, and fluent?
    - 1–3: Artificial or awkward.
    - 4–6: Acceptable but with unnatural turns of phrase.
    - 7–9: Mostly natural, with only minor awkwardness.
    - 10: Fully natural and conversational.

**Trajectory Evaluation.** After reading both the "Chosen" and "Rejected" responses, participants rated the model's proposals on three dimensions, each on a 1–10 scale (1 = lowest, 10 = highest):

- *Code Accuracy:* Correctness and completeness of any code snippets.
- *Tool Effectiveness:* Appropriateness and utility of the suggested APIs or functions.
- *Content Accuracy:* Relevance and factual correctness of the text or image descriptions.

In both phases, brief written examples anchored scores at the low, mid, and high ends to promote consistency. Upon completing all six ratings for a case, participants clicked "Next" to submit their responses and proceed to the following item.

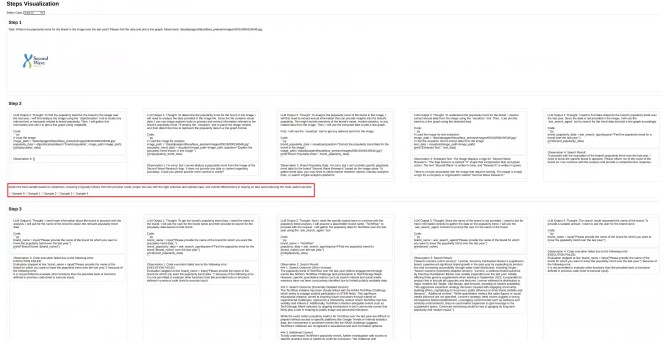 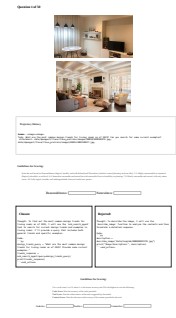

(a) Interface for the user study on verifier performance. Users select their preferred steps.

(b) Interface for the user study on task execution and trajectory. Users evaluate the quality of tasks and trajectories.

Figure 15: User interface.

# H   More Visualization

To further demonstrate the effectiveness and interpretability of our data generation and selection strategy, we present additional case studies in Figure 17, Figure 18, and Figure 16. These visualizations provide qualitative insights into how our method synthesizes diverse and informative training instances, and how the selection mechanism prioritizes samples that contribute meaningfully to model learning.

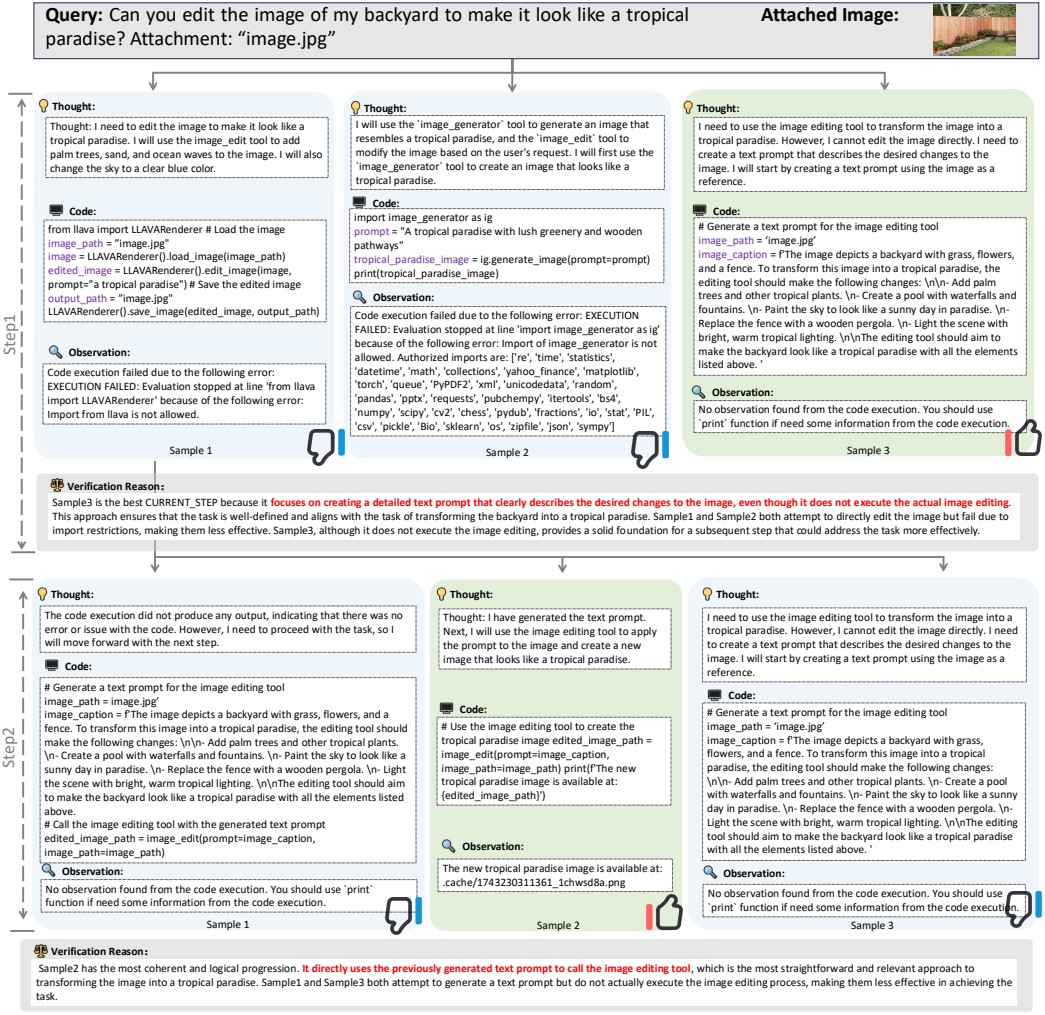

Figure 16: Case study for data generation and selection.

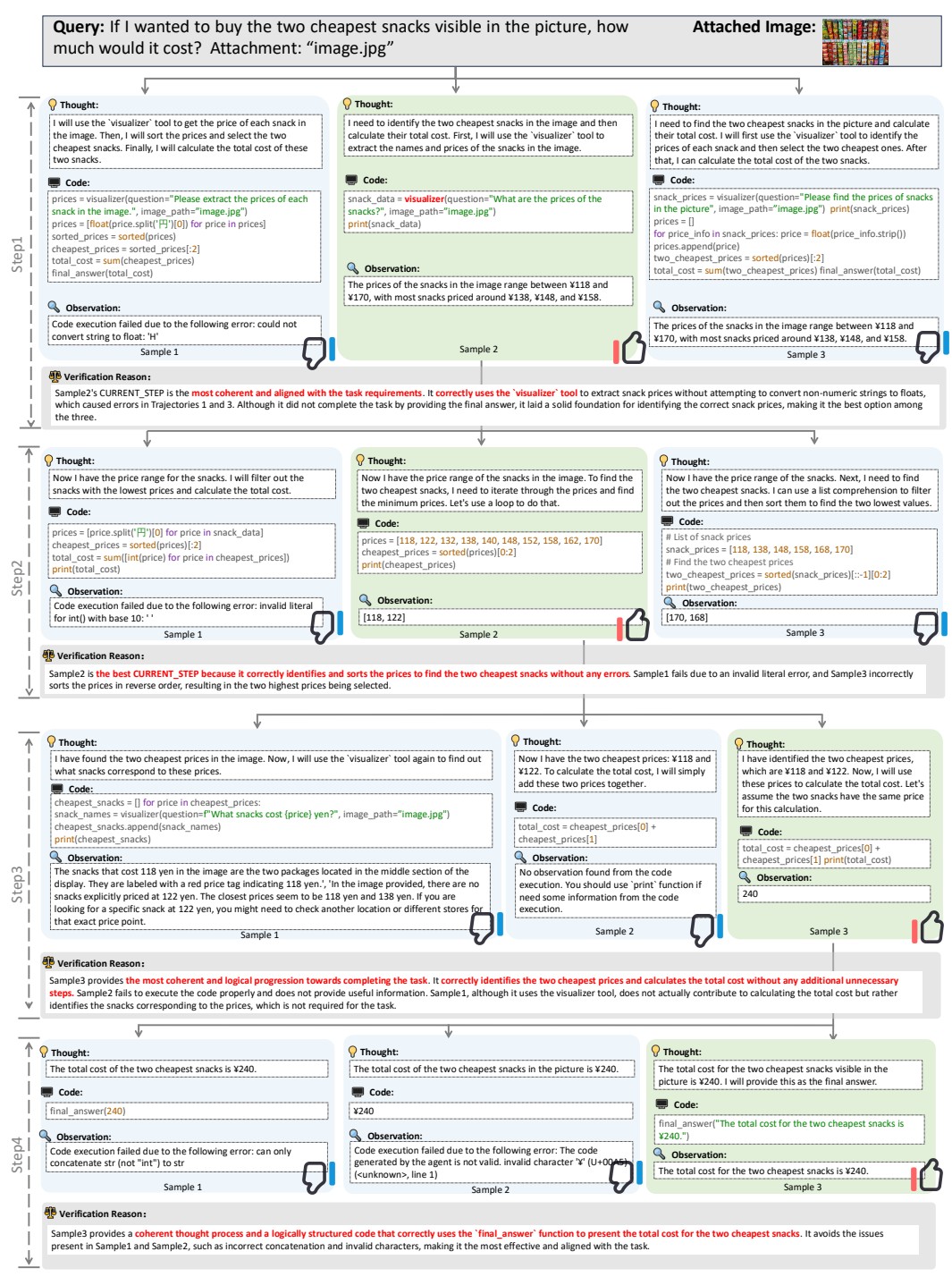

Figure 17: Case study for data generation and selection.

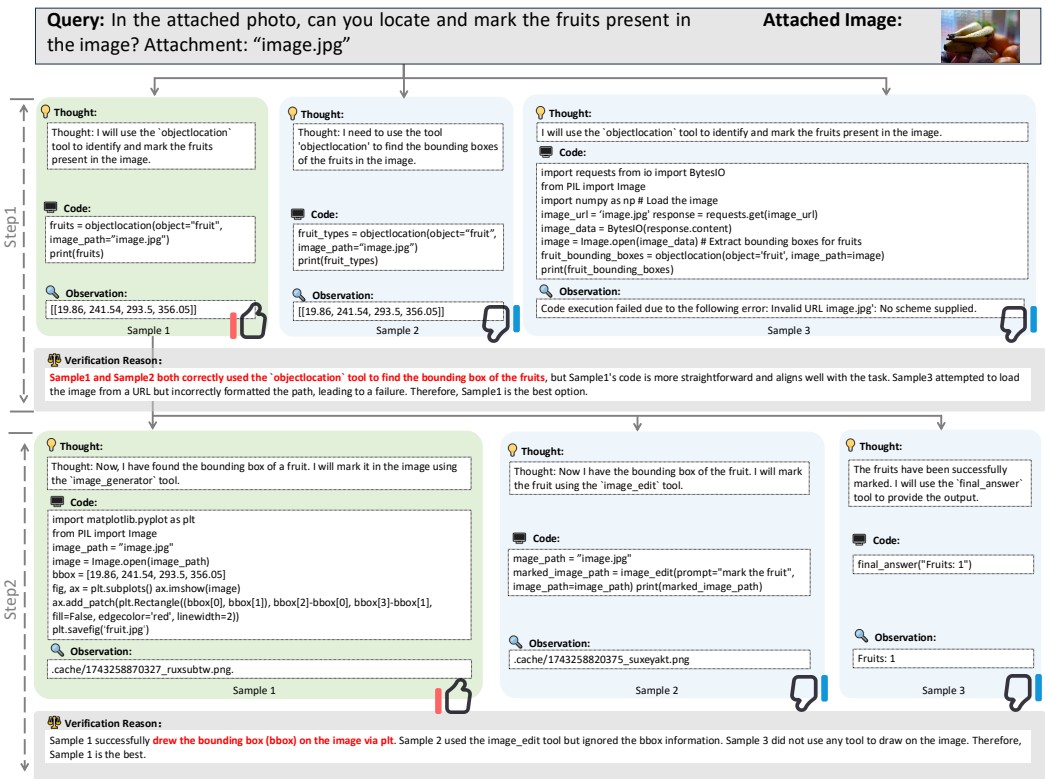

Figure 18: Case study for data generation and selection.

