# OpenReview forum: "Iterative Tool Usage Exploration for Multimodal Agents via Step-wise Preference Tuning"
_NeurIPS.cc/2025/Conference — NeurIPS 2025 poster_

### Official Review · Reviewer_Zye7 · 2025-06-23

**Clarity:** 3
**Significance:** 3
**Originality:** 3
**Rating:** 5
**Confidence:** 4

**Summary:**

This paper introduces SPORT (Step-wise Preference Optimization for Refining Trajectories), a novel method for training multimodal agents without requiring pre-collected human annotations. Current methods for training multimodal agents rely heavily on expensive human-annotated preference pair data or distilled tool usage trajectories data from closed-sourced APIs, which creates a bottleneck for developing agents that can handle complex multimodal tasks. SPORT enables agents to automatically improve their tool usage capability through self exploration and iterative refinement. This method consists of four iterative components:

1. **Task generation**: synthesizes diverse multimodal tasks (queries + files) using LLMs.
2. **Step sampling**: the agents explores multiple possible actions at each step by sampling different tool calls.
3. **Step verification**: a verifier evaluates the sampled actions to identify the preferred option for each step.
4. **Preference tuning**: uses the collected step-wise preference data to update the agent controller via DPO.

**Main contributions**:

* A self-exploration framework that eliminates the need for pre-collected annotations.
* Step-wise preference optimization enables improvements over SFT baselines on GTA and GAIA benchmarks.
* A dataset of 16K preference data samples is created through the exploration process.

Through multiple ablation experiments and user study, this paper demonstrates that agents can effectively learn to use tools through autonomous exploration, which provides an alternative to traditional supervised approaches.

**Questions:**

1. **Task seed collection and diversity**:
The paper says "we collect seed from the existing method MAT" but doesn't clarify the specific source or scale. Could you please clarify the following: Did you sample seed tasks directly from the 20K MM-Traj dataset, or did you use the MAT framework to generate new seed tasks? How many seed tasks were used in total for task synthesis? And lastly, how did you ensure diversity across task categories? (e.g., did you sample uniformly across MAT's 16 non-overlapping categories mentioned in their paper?)
2. **Verifier details**:
 What model(s) do you use for the verifier? How did you decide on the selected verifier or evaluate the verifier? Your appendix has a section for the prompt for step verifier, but it would be helpful to include more technical details and some analysis in the main section (e.g. Section 3.3).
3. **Experimental baselines and comparisons**:
It’s great that you list the GTA and GAIA results for many combinations of methods and controllers. Do you also have the GTA and GAIA results for the SPORT Agent with the initial agent controller (the post-SFT model, i.e., the MAT-Qwen2-VL-7B)? This might create a more direct comparison showing SPORT's improvement over the same base model. Also, did you conduct experiments with your agent using closed-source controllers? I'm curious if with the same closed-source model, your agent would perform better than other agents (e.g. HFAgent).
4. **Efficiency and cost analysis**:
I think it might be helpful to have some discussion about the compute time and computational costs. For example, a high-level comparison between SPORT and conventional methods. If the gains are much larger than the overhead, this would strengthen the contribution.

**Ethical Concerns:**

["NO or VERY MINOR ethics concerns only"]

**Final Justification:**

After reviewing the author's rebuttal and our discussion, I maintain my recommendation to Accept this paper.

Resolved Issues:
- Verifier details: Authors clearly explained using Qwen2.5-7B as verifier, with ablation results showing 60.26% vs 55.13% accuracy on GTA compared to using Qwen2-VL-7B. This choice makes sense.
- Computational costs: The efficiency analysis shows SPORT actually uses less compute than the MAT baseline (82 vs 122 GPU hours), which strengthens the contribution.
- Task diversity: Clarified use of 425 human-written seeds with random sampling for diversity—this is reasonable.

Remaining Limitations (Acceptable):
- Performance gap with closed-source models: While SPORT underperforms on complex benchmarks like GAIA, this is due to model scale limitations rather than the method itself. Given the focus on open-source accessibility, this is understandable.
- Building on existing work: Though not architecturally novel, the step-wise preference optimization genuinely addresses an important problem.

Key Strengths: The paper tackles a real bottleneck (expensive human annotations for agent training) with a practical solution. The experimental validation is thorough (including 82% human-verifier alignment), and the method works with accessible open-source models. The generated 16K preference dataset will benefit future research.

Bottom Line: This is a technically solid paper with clear practical value. The step-wise preference optimization is well-executed and addresses a genuine need in the multimodal agent community. The authors have satisfactorily addressed my concerns, and the remaining limitations don't diminish the core contribution.

**Limitations:**

yes

**Paper Formatting Concerns:**

The paper is well-formatted overall. A few minor editorial suggestions:

1. In Section 1 Introduction (line 37): “Based on the above observation, we expect that the agent automatically generates tasks” - should be "expect that agents automatically" (plural consistency).
2. In Section 5 Conclusion (line 354): "causing inferior generalization for some outlines" - "outlines" seems like a typo, possibly meant "outliers".
3. In Appendix C Task Generation (line 559): “we first use an LLM to generate queres” - "queres" seems like a typo, possibly meant "queries".

**Quality:**

3

**Strengths And Weaknesses:**

## Quality
**Strengths**:
* Technical merit: The paper presents a well-design iterative framework with clear technical components. Its step-wise preference optimization approach is theoretically grounded for DPO.
* Experimental validation: The method is experimented on two popular benchmarks (GTA and GAIA) with extensive ablation studies. The authors also conduct detailed statistic analysis on the generated preference data in three aspects (code error rate, tool selection, and content variations).
* Human validation: The work has thorough validation including user studies to validate both data quality and verifier alignment with human preferences (82% agreement rate).

**Weaknesses**:
* To my understanding, the verifier component seems to be an important element in this system. The paper would benefit from more details and analysis of its implementation and impact on performance.

## Clarity
**Strengths**:
* Well-structured presentation: The paper follows a logical flow from problem motivation through method description to experimental validation.
* Clear visualizations: The figures and tables in the paper effectively illustrate the pipeline (Figure 1), search scheme (Figure 2), concrete examples of preference data construction (Figure 4 and 5), and training process through algorithm.
* Clear communication and replication support: The authors presents technical concepts with clear and accessible language while providing comprehensive implementation details about model, training, and prompt templates.

**Weaknesses**:
* Verifier design: While prompt templates are included in the appendix, the main paper sections could better elaborate on the verifier's design choices and implementation details. A more thorough discussion of the verifier's architecture, design rationale, and key parameters would enhance the paper's technical depth.

## Significance
**Strengths**:
* SPORT removes the dependency on expensive human annotations for multimodal agent training, which solves a key practical challenge.
* This method works with existing open-source models (Qwen2-VL-7B) and standard toolkits, making it accessible to the broader community.
* The 16K preference dataset generated will be valuable for future experiments and research.

**Weaknesses**:
* Future performance potential: While the experiments show promising results with open-source models, it has not yet matched closed-source model based agents' performance. This gap suggests potential for future improvements.

## Originality
**Strengths**:
* The idea of constructing preference data at the step level rather than trajectory level is innovative and allows learning from partial successes.
* The system effectively combines task synthesis, online exploration, and preference learning in a novel way for multimodal settings.

**Weaknesses**:
* The work seems effectively build upon and enhances the MAT framework and MM-Traj dataset. Although effective, more original architectural contributions could further strengthen the work's novelty.

---

> ### Author Rebuttal · Authors · 2025-07-31
>
> Thank you for your comprehensive review. We’re especially grateful for your recognition of the technical design, experimental depth, and practical impact of our step-wise preference optimization framework. To streamline our response, we denote each Weakness as **W**, each Question as **Q**, and our Answer as **A**. For brevity, Qwen2-VL-7B and Qwen2.5-7B refer to their Instruct versions. We address your concerns one by one below.
>
> > **W1** To my understanding, the verifier component seems to be an important element in this system. The paper would benefit from more details and analysis of its implementation and impact on performance.
>
> > **W2.** Verifier design: While prompt templates are included in the appendix, the main paper sections could better elaborate on the verifier's design choices and implementation details. A more thorough discussion of the verifier's architecture, design rationale, and key parameters would enhance the paper's technical depth.
>
> > **Q2.** Verifier details: What model(s) do you use for the verifier? How did you decide on the selected verifier or evaluate the verifier? Your appendix has a section for the prompt for step verifier, but it would be helpful to include more technical details and some analysis in the main section (e.g. Section 3.3).
>
> **A.** Thanks for your suggestion. We prioritized open-source models to ensure reproducibility and facilitate future research.   We selected Qwen2.5-7B as the verifier because qualitative analysis showed it performed better in evaluating code accuracy and tool selection appropriateness compared to Qwen2-VL-7B. We attribute this to Qwen2-VL-7B's image inputs occupying significant context window space, limiting the ability for verification. Quantitative experiments confirmed this: re-generating preference annotations for using Qwen2-VL-7B as verifier yielded 55.13% accuracy on GTA benchmark, while Qwen2.5-7B achieved 60.26% (Table A), validating our choice.
>
> **Table A.** Ablation about AI verifier, accuracies (%) on GTA benchmark.
> | Baseline | Qwen2-VL-7B as Verifier | Qwen2.5-7B as Verifier |
> |-|-|-|
> | 53.85| 55.13 |60.26|
>
> Additionally, our human evaluation (lines 324–330 and Table 5) shows that Qwen2.5-7B achieves an 82% alignment rate in step verification with human participants, confirming the effectiveness of our design choice.
>
> > **W3.** Future performance potential: While the experiments show promising results with open-source models, it has not yet matched closed-source model based agents' performance. This gap suggests potential for future improvements.
>
> **A.** Thank you for this thoughtful suggestion. We acknowledge there is substantial room for future improvement. While the SPORT Agent demonstrates strong performance on GTA with open-source models, it underperforms closed-source based agents on the more complex GAIA benchmark. This underperformance stems primarily from the limited parameter scale of open-source base models, which constrains their reasoning capabilities. In contrast, closed-source models like GPT-4o achieve superior performance through their larger model parameters, more extensive training data, and longer context windows, enabling enhanced reasoning abilities. We plan to explore stronger open-source backbones and improved verifier generalization in future work.
>
> > **W4.** The work seems effectively build upon and enhances the MAT framework and MM-Traj dataset. Although effective, more original architectural contributions could further strengthen the work's novelty.
>
> **A.** Thank you for the valuable feedback. Our contribution primarily lies in the design of an iterative tool usage exploration framework for multimodal agents. Specifically, we address an orthogonal yet important challenge to architectural design: *enabling agents to autonomously discover effective tool usage strategies through self-exploration, especially in the absence of ground-truth supervision*. We agree that architectural innovations are also vital for advancing agent capabilities, and we look forward to exploring this direction in future work to further enhance the capabilities of agents.
>
> > **Q1.** Task seed collection and diversity: The paper says "we collect seed from the existing method MAT" but doesn't clarify the specific source or scale. Could you please clarify the following: Did you sample seed tasks directly from the 20K MM-Traj dataset, or did you use the MAT framework to generate new seed tasks? How many seed tasks were used in total for task synthesis? And lastly, how did you ensure diversity across task categories? (e.g., did you sample uniformly across MAT's 16 non-overlapping categories mentioned in their paper?)
>
> **A.** We directly adopted the 425 human-written task seeds from the task generation process in MAT [b], as these seeds offer high quality via their manual curation. To ensure diversity in the generated tasks, we randomly sample and shuffle 20 seeds as in-context examples for each query generation. Additionally, when generating tasks, we provide the complete toolset descriptions to the model to facilitate more reasonable and diverse task outputs.
>
>
> [b] Multi-modal Agent Tuning: Building a VLM-Driven Agent for Efficient Tool Usage.
>
> > **Q3.** Experimental baselines and comparisons: It’s great that you list the GTA and GAIA results for many combinations of methods and controllers. Do you also have the GTA and GAIA results for the SPORT Agent with the initial agent controller (the post-SFT model, i.e., the MAT-Qwen2-VL-7B)? This might create a more direct comparison showing SPORT's improvement over the same base model. Also, did you conduct experiments with your agent using closed-source controllers? I'm curious if with the same closed-source model, your agent would perform better than other agents (e.g. HFAgent).
>
> **A.** To ensure a fair comparison, the SPORT Agent uses the same toolset (lines 212-213 in the manuscript) and agent architecture as the T3-Agent and HF Agent. So the results in Table 1 already provide a direct comparison among MAT-Qwen2-VL-7B (using T3-Agent), original Qwen2-VL-7B (using HF Agent), closed-source models (using HF Agent), and our method, where only the controllers are different. As shown in Table 1, our controller already has comparable, and even better performance than closed-source models in the GTA benchmark, but worse performance in the GAIA benchmark, mainly due to the complexity of GAIA. We plan to explore stronger open-source backbones in future work.
>
> > **Q4.** Efficiency and cost analysis: I think it might be helpful to have some discussion about the compute time and computational costs. For example, a high-level comparison between SPORT and conventional methods. If the gains are much larger than the overhead, this would strengthen the contribution.
>
>
> **A.** Thank you for the insightful suggestion. We add a high-level comparison of compute time and GPU usage between the baseline method (MAT) and our SPORT framework. Table F summarizes the estimated compute time and GPU cost for both methods.
>
> The computational cost of *Task Synthesis*, *Step Sampling*, and *Tool Calling* is closely tied to the data scale used in the data generation stage. MAT synthesizes and samples steps for 20K tasks and 20K trajectories, resulting in high overhead across all three components. SPORT performs Task Synthesis on 2K tasks and constructs 16K preference pairs for tuning, leading to reduced compute time in these stages.
>
> In addition, SPORT introduces an extra Step Verification process to obtain step-level preferences. While this stage incurs some additional costs, it remains acceptable relative to the total runtime due to the smaller data volume.
>
>
> **Table F.** Estimated compute time and GPU cost comparison between MAT and SPORT. GPU hours (GPUh) are measured as usage of a single A100 80GB GPU for one hour. The estimates for MAT are derived from its reported tool invocation frequency, combined with empirical measurements of the GPT-4o-mini API and tool execution latency on our hardware.
>
>
> | Component| MAT Time (h) | SPORT Time (h) | MAT Cost (GPUh) | SPORT Cost (GPUh) |
> |-|-|-|-|-|
> |***Data Generation***|20K tasks, 20K trajectories |2K tasks, 16K perference data |||
> |Task Synthesis | 29.13| 3.61 | 0 (GPT-4o-mini API ~500\$)| 3.61 (Qwen2-VL-7B)|
> |Step Sampling| 69.92 | 4.53| 0 (GPT-4o-mini API  ~1500\$)  | 4.53 (Tuned-Qwen2-VL-7B)|
> |Step Verification| 0 | 3.36| 0 | 3.36 (Qwen2.5-7B)|
> | Tool Calling | 58.93 | 9.86 | 58.93|9.86 |
> | ***Training*** | |  |  | |
> | Model Training w/ 4$\times$ A100 80G GPU|15.77|15.20| 63.08| 60.80 |
> | **Total**|173.75 |36.56|122.01|82.16|
>
>
> > **Paper Formatting Concerns** The paper is well-formatted overall. A few minor editorial suggestions:
> > 1. In Section 1 Introduction (line 37): “Based on the above observation, we expect that the agent automatically generates tasks” - should be "expect that agents automatically" (plural consistency).
> > 2. In Section 5 Conclusion (line 354): "causing inferior generalization for some outlines" - "outlines" seems like a typo, possibly meant "outliers".
> > 3. In Appendix C Task Generation (line 559): “we first use an LLM to generate queres” - "queres" seems like a typo, possibly meant "queries".
>
> **A.** We sincerely thank the reviewer for the careful reading and for pointing out these typos. We will correct all of them in the revised version and perform a thorough proofreading pass to ensure overall consistency and clarity in the manuscript.

---

> > ### Comment · Reviewer_Zye7 · 2025-08-05
> >
> > Thank you for the thorough response to my comments! The clarifications provided have adequately addressed my concerns. I'll maintain my positive score.

---

> > > ### Author Response · Authors · 2025-08-06
> > > **Thanks for your review!**
> > >
> > > Thank you so much for your thoughtful response, as well as the time and effort you dedicated to reviewing both our manuscript and rebuttal. We’re very glad our responses addressed your concerns and truly appreciate your constructive engagement!

---

### Official Review · Reviewer_avvW · 2025-06-29

**Clarity:** 3
**Significance:** 2
**Originality:** 2
**Rating:** 4
**Confidence:** 2

**Summary:**

This paper presents SPORT, a framework for training multimodal tool-use agents using step-wise preference optimization. SPORT builds its own preference data via self-exploration: at each decision step, it samples multiple tool-usage actions, executes them, and uses an LLM-based verifier to select the most desirable one. Crucially, the verifier compares outcomes in terms of usefulness rather than correctness, providing more flexible supervision. The authors evaluate SPORT on two multimodal benchmarks: GTA (image-based reasoning) and GAIA (document-based tasks).

**Questions:**

- How sensitive is SPORT to the quality and diversity of the synthetic tasks generated in the early stages of self-exploration?

**Ethical Concerns:**

["NO or VERY MINOR ethics concerns only"]

**Final Justification:**

The authors provide additional experiments, which help address some of my concerns. My only remaining concern is still W1, regarding the novelty of the work. However, I'm not an expert in this specific domain, so I maintain a low confidence score.

**Limitations:**

yes

**Quality:**

3

**Strengths And Weaknesses:**

### Strengths
- SPORT demonstrates clear improvements over SFT baselines and performs competitively with agents built on top of strong models like GPT-4o.
- The full training pipeline, including task generation, tool sampling, and preference labeling, is entirely self-supervised, requiring no human-annotated trajectories.

### Weaknesses
- The main components of SPORT, including self-instruct task generation, LLM-based comparison, and DPO, are all established techniques. While the paper positions itself as "explores the tool usage by itself via an iterative manner", it does not introduce any novel mechanisms for agent-level self-improvement, such as strategic planning, error reflection, or hierarchical reasoning. Instead, it looks more like a specific application of existing methods to tool-usage domain with tool-based actions as action space. Due to the limited novelty, I put a low confidence score.
- While very relevant related works such as Step-DPO are discussed, it is not included in experimental comparisons. A fair adaptation of such methods to the tool-use setting would strengthen the empirical claims. For example an adapted version of Step-DPO that does error localization and correction.

---

> ### Author Rebuttal · Authors · 2025-07-31
>
> Thank you for your insightful and constructive feedback. We appreciate your recognition of SPORT's fully self-supervised training pipeline and its ability to perform competitively with stronger models like GPT-4o, all without any human-annotated trajectories. For simplicity, we mark each Weakness as **W**, each Question as **Q**, and our Answer as **A**. Throughout this response, Qwen2-VL-7B and Qwen2.5-7B refer to the Instruct versions. We address your comments one by one below.
>
> > **W1.** The main components of SPORT, including self-instruct task generation, LLM-based comparison, and DPO, are all established techniques. While the paper positions itself as "explores the tool usage by itself via an iterative manner", it does not introduce any novel mechanisms for agent-level self-improvement, such as strategic planning, error reflection, or hierarchical reasoning. Instead, it looks more like a specific application of existing methods to tool-usage domain with tool-based actions as action space. Due to the limited novelty, I put a low confidence score.
>
> **A.** We appreciate the reviewer’s feedback. SPORT addresses a different but important challenge compared with agent mechanisms: *how can agents autonomously discover effective tool usage strategies in multimodal scenarios through self-exploration when no ground truth exists?*
>
> Our key contribution is **a tool usage exploration framework** for multimodal agents, which integrates task synthesis, parallel step sampling, step verification, and DPO updates. In this case, the agent could learn tool usage without any pre-collected data. While based on existing components, SPORT repurposes them into a new training paradigm that supports scalable self-improvement without manual supervision. SPORT deliberately focuses on learning tool usage primitives by itself, which is **orthogonal to agent abilities like strategic planning, error reflection, or hierarchical reasoning**. These abilities are not in conflict with our goal and can be naturally incorporated into our framework to further enhance agent capabilities.
>
> > **W2.** While very relevant related works such as Step-DPO are discussed, it is not included in experimental comparisons. A fair adaptation of such methods to the tool-use setting would strengthen the empirical claims. For example an adapted version of Step-DPO that does error localization and correction.
>
> **A.** Thank you for the suggestion. We have conducted experiments comparing our method with an adapted variant of Step-DPO[a] for the multimodal agent setting. Specifically, we use GPT-4o-mini to sample positive step-level responses for the same tasks used in SPORT, while responses from MAT-Qwen2-VL-7B serve as negative examples. This allows us to construct step-level preference pairs for DPO training. The resulting model, MAT-Qwen2-VL-7B-DPO, shows modest improvements over the SFT baseline (e.g., +1.28% AnsAcc, +2.67% ToolAcc), as shown in Table 3. In contrast, our SPORT method achieves larger gains across all metrics, demonstrating the effectiveness of our adaptive self-exploration framework under unlabeled multimodal settings.
>
> **Table 3.** Ablation for MAT-SFT vs. MAT-SFT-DPO vs. SPORT on the GTA benchmark.
> |Method|AnsAcc (%)|ToolAcc (%)|CodeExec (%)|
> |-|-|-|-|
> |MAT-Qwen2-VL-7B |53.85|64.63|84.32|
> |MAT-Qwen2-VL-7B-DPO|55.13|67.30|85.90|
> |SPORT (Ours) |60.26|72.41|91.87|
>
> [a] Step-DPO: Step-wise Preference Optimization for Long-chain Reasoning of LLMs.
>
> > **Q1.** How sensitive is SPORT to the quality and diversity of the synthetic tasks generated in the early stages of self-exploration?
>
>
> **A.** Task quality and diversity in early stages do have an impact on SPORT’s performance. We conducted an ablation for task synthesis, where the in-context examples were reduced from 20 to 5 and the task seed pool was narrowed from 425 to 100. This setup was designed to mimic a low-diversity scenario in early-stage exploration. As shown in Table E, this led to a moderate drop in performance (60.26 → 58.33). However, the result still significantly outperforms the SFT baseline (58.33 vs. 53.85).
>
>
> **Table E.** Impact of task diversity on the GTA benchmark.
> |Method|AnsAcc (%)|
> |-|-|
> |MAT-Qwen2-VL-7B |53.85|
> |SPORT w/ 5 from 100 seeds| 58.33 |
> |SPORT  w/ 20 from 425 seeds|60.26|

---

> > ### Comment · Reviewer_avvW · 2025-08-04
> >
> > Thank you for the additional experiments, which help address some of my concerns. I would like to maintain my original positive score. Regarding weakness w1, I still hold my original opinion, though I acknowledge that my confidence on this point is low.

---

> > > ### Author Response · Authors · 2025-08-05
> > > **Thanks for your review!**
> > >
> > > Thank you for your positive assessment and for maintaining your positive score. We truly appreciate your thoughtful engagement throughout the review process. We understand and respect your perspective regarding the novelty aspect, and we’re glad the additional experiments helped address your other concerns. Your feedback raised important points and contributed meaningfully to our work.

---

### Official Review · Reviewer_rUdq · 2025-07-02

**Clarity:** 3
**Significance:** 3
**Originality:** 3
**Rating:** 4
**Confidence:** 1

**Summary:**

This paper presents SPORT (step-wise preference optimization to refine the trajectories of tool usage), a framework for training multimodal agents to use tools effectively without requiring pre-collected human annotations. Current multimodal agent training relies heavily on expensive human-annotated trajectories and tool usage data. This creates bottlenecks and may lead to poor generalization due to distribution mismatch with target environments. On the contrary, SPORT operates without human annotations through four iterative components: 1) Task Synthesis: Uses LLMs to generate diverse multimodal tasks (queries + files). 2) Step Sampling: Agent samples multiple candidate actions for each step. 3) Step Verification: AI verifier evaluates and ranks step-level actions based on quality. 4) Preference Tuning: Uses Direct Preference Optimization (DPO) to fine-tune the agent controller.

Results on GTA and GAIA benchmarks show that SPORT achieves 6.41% and 3.64% improvements over supervised fine-tuning baselines, demonstrating effectiveness in multimodal reasoning tasks.

**Questions:**

1. Computational cost: Seems like the proposed method requires expensive tool execution (LLM calls, web searches) during exploration. The reviewer hope the author could provide more analysis on computational cost estimation and comparison.

2. Selection of the LLM model in task synthesis and AI verifier. The reviewer is wondering how the LLM model and AI verifier selection would influence the final results. It seems that in the first three steps, all the knowledge comes from these two model (and maybe the unlabeled data). However, there is no ablation study on what influences these two models bring in the whole framework. It would be valuable to provide some experiment results on this.

3. Influence of unlabeled data used in the training. It would be better to provide experiment results on using different portion of unlabeled data for training the framework.

**Ethical Concerns:**

["NO or VERY MINOR ethics concerns only"]

**Final Justification:**

The reviewer's concerns are resolved by the response. The reviewer is willing to maintain the positive score.

**Limitations:**

1. Computational cost: Seems like the proposed method requires expensive tool execution (LLM calls, web searches) during exploration. The reviewer hope the author could provide more analysis on computational cost estimation and comparison.

2. Selection of the LLM model in task synthesis and AI verifier. The reviewer is wondering how the LLM model and AI verifier selection would influence the final results. It seems that in the first three steps, all the knowledge comes from these two model (and maybe the unlabeled data). However, there is no ablation study on what influences these two models bring in the whole framework. It would be valuable to provide some experiment results on this.

3. Influence of unlabeled data used in the training. It would be better to provide experiment results on using different portion of unlabeled data for training the framework.

**Paper Formatting Concerns:**

N.A.

**Quality:**

3

**Strengths And Weaknesses:**

1. Self-Improving Framework: The iterative nature allows agents to continuously improve through interaction with real environments
2. Evaluation: Tests on two established benchmarks (GTA, GAIA).
3. Practical Design: The proposed method works with open-source models (Qwen2-VL-7B), generates diverse synthetic tasks covering multiple file types and achieves competitive performance against closed-source models.

---

> ### Author Rebuttal · Authors · 2025-07-31
>
> Thank you for your insightful and constructive feedback. We’re glad you found SPORT’s annotation-free, self-improving framework practical and effective, particularly its ability to scale using only open-source models and synthetic tasks. To make our replies easier to follow, we label each Question as **Q**, and our corresponding Answer as **A**. All references to Qwen2-VL-7B and Qwen2.5-7B denote their Instruct versions. We address your comments one by one below.
>
>
> > **Q1.** Computational cost: Seems like the proposed method requires expensive tool execution (LLM calls, web searches) during exploration. The reviewer hope the author could provide more analysis on computational cost estimation and comparison.
>
>
> **A.** Thanks for your suggestion. We present a detailed analysis of the computational cost and time required for sampling and training on 500 tasks within the SPORT framework, summarized in Table C.
>
> Table C encompasses time for task synthesis, tool calling, model inference, and model updating. Notably, tool calling, particularly web searches via the Google Search API, accounts for a significant portion of the time due to real-time external interactions. We will incorporate this detailed computational cost analysis into the revised manuscript.
>
> **Table C.** Time and computation cost estimation for each stage in the SPORT framework for 500 tasks.
> |Stage|Model|Time(mins)|Time Proportion|Config|
> |-|-|-|-|-|
> |Task Synthesis (Inference)|Qwen2-VL-7B|54.18|12.47%|1$\times$A100 80G GPU|
> |Step Sampling (Inference)|Tuned-Qwen2-VL-7B|68.02|15.65%|1$\times$A100 80G GPU|
> |Step Verification (Inference)|Qwen2.5-7B|50.45|11.61%|1$\times$A100 80G GPU|
> |Tool calling|Tool Models + Google Search API|147.87|34.04%| 1$\times$A100 80G GPU|
> |DPO Training|Tuned-Qwen2-VL-7B|114.00|26.23%|8$\times$A100 80G GPU|
> |Total|N/A|434.53|100.00% | ~20 A100 GPU Hours Total |
>
> > **Q2.** Selection of the LLM model in task synthesis and AI verifier. The reviewer is wondering how the LLM model and AI verifier selection would influence the final results. It seems that in the first three steps, all the knowledge comes from these two model (and maybe the unlabeled data). However, there is no ablation study on what influences these two models bring in the whole framework. It would be valuable to provide some experiment results on this.
>
> **A.** We thank the reviewer for this valuable question. Here we provide more experimental results and analysis about this.
>
> **Task Synthesis:** We chose Qwen2-VL-7B for task synthesis. To assess sensitivity to this choice, we conducted a human study (detailed in Appendix C.1), comparing task quality generated by Qwen2-VL-7B versus GPT-4o-mini across Naturalness and Reasonableness dimensions. As shown in Table 7, both models produced tasks of comparable quality, demonstrating that Qwen2-VL-7B is robust and capable of high-quality task generation.
>
> **Table 7.** User study for open-source vs. closed-source models generated tasks. Scores are scaled from 1 to 10, and a higher score denotes better quality.
>
> | Model| Task Naturalness | Task Reasonableness |
> |-|-|-|
> | GPT-4o-mini|8.71|8.37|
> | QWen2-VL-7B|8.75|8.35|
>
> **AI Verifier.** We prioritized open-source models to ensure reproducibility and facilitate future research. We selected Qwen2.5-7B as the verifier because qualitative analysis showed it performed better in evaluating code accuracy and tool selection appropriateness compared to Qwen2-VL-7B. We attribute this to Qwen2-VL-7B's image inputs occupying significant context window space, limiting available context for verifier instructions. Quantitative experiments confirmed this: re-generating preference annotations for using Qwen2-VL-7B as verifier yielded 55.13% accuracy on the GTA benchmark, while Qwen2.5-7B achieved 60.26% (Table A), validating our choice.
>
> **Table A.** Ablation about AI verifier, accuracies (%) on GTA benchmark.
> | Baseline | Qwen2-VL-7B as Verifier | Qwen2.5-7B as Verifier |
> |-|-|-|
> | 53.85| 55.13 |60.26|
>
> Additionally, our human evaluation (lines 324–330 and Table 5) shows that Qwen2.5-7B achieves an 82% alignment rate in step verification with human participants, confirming the effectiveness of our design choice.
>
> > **Q3.** Influence of unlabeled data used in the training. It would be better to provide experiment results on using different portions of unlabeled data for training the framework.
>
> **A.** Thank you for your valuable suggestion. We have added ablation experiments using different portions of unlabeled data for training. The experimental results are shown in Table D. The results show that SPORT's performance improves consistently with more unlabeled data, achieving gains of 1.92% (4K→8K) and 2.57% (8K→16K). This indicates that our method can effectively utilize self-generated preference data to improve tool usage capabilities.
>
> **Table D.** Answer accuracies (AnsAcc) on GTA benchmark with varying amounts of unlabeled data used in DPO training.
> |Unlabeled data num|AnsAcc (%)|
> |-|-|
> |4K (25%)|55.77|
> |8K (50%)|57.69|
> |16K (100%)|60.26|

---

> > ### Comment · Reviewer_rUdq · 2025-08-05
> >
> > Thank you for the detailed response. The concerns about Q2 and Q3 are resolved by the new experiments. For Q1, is it possible to provide efficiency comparison to other baseline methods?

---

> > > ### Author Response · Authors · 2025-08-06
> > > **Response to Reviewer rUdq**
> > >
> > > Thank you for the thoughtful follow-up! We’re glad to hear that the additional experiments have addressed your concerns regarding Q2 and Q3. To further clarify the computational efficiency aspect raised in Q1, we provide a detailed comparison between SPORT and the baseline method MAT [a] as shown in Table F.
> > >
> > > **Table F.** Estimated compute time and GPU cost comparison between MAT and SPORT. GPU hours (GPUh) are measured as usage of a single A100 80GB GPU for one hour. The estimates for MAT are derived from its reported tool invocation frequency, combined with empirical measurements of the GPT-4o-mini API and tool execution latency on our hardware.
> > >
> > >
> > > | Component| MAT Time (h) | SPORT Time (h) | MAT Cost (GPUh) | SPORT Cost (GPUh) |
> > > |-|-|-|-|-|
> > > |***Data Generation***|20K tasks, 20K trajectories |2K tasks, 16K perference data |||
> > > |Task Synthesis | 29.13| 3.61 | 0 (GPT-4o-mini API ~500\$)| 3.61 (Qwen2-VL-7B)|
> > > |Step Sampling| 69.92 | 4.53| 0 (GPT-4o-mini API  ~1500\$)  | 4.53 (Tuned-Qwen2-VL-7B)|
> > > |Step Verification| 0 | 3.36| 0 | 3.36 (Qwen2.5-7B)|
> > > | Tool Calling | 58.93 | 9.86 | 58.93|9.86 |
> > > | ***Training*** | |  |  | |
> > > | Model Training w/ 4$\times$ A100 80G GPU|15.77|15.20| 63.08| 60.80 |
> > > | **Total**|173.75 |36.56|122.01|82.16|
> > >
> > >
> > > The computational cost of *Task Synthesis*, *Step Sampling*, and *Tool Calling* is closely tied to the data scale used in the data generation stage. MAT synthesizes and samples steps for 20K tasks and 20K trajectories, resulting in high overhead across all three components. SPORT performs Task Synthesis on 2K tasks and constructs 16K preference pairs for tuning, leading to reduced compute time in these stages.
> > >
> > > In addition, SPORT introduces an extra Step Verification process to obtain step-level preferences. While this stage incurs some additional costs, it remains acceptable relative to the total runtime due to the smaller data volume.
> > >
> > > We will incorporate this comparison and discussion into the revised version of the paper. We thank the reviewer again for the thoughtful feedback.
> > >
> > > [a] Multi-modal Agent Tuning: Building a VLM-Driven Agent for Efficient Tool Usage. ICLR 2025.

---

> ### Comment · Reviewer_rUdq · 2025-08-06
>
> Thank you for the detailed response. The reviewer's concerns are resolved and would like to maintain the positive score.

---

> > ### Author Response · Authors · 2025-08-07
> > **Thanks for your review!**
> >
> > Thank you for your thoughtful comments and follow-up! We're glad that our additional experiments addressed your concerns. Your suggestions were very helpful in improving the clarity and completeness of our work, and we sincerely appreciate your constructive engagement throughout the review process.

---

### Official Review · Reviewer_VNFD · 2025-07-03

**Clarity:** 2
**Significance:** 2
**Originality:** 3
**Rating:** 4
**Confidence:** 3

**Summary:**

This paper introduces SPORT, an iterative self-exploration framework for training multimodal agents in tool usage without any annotated data. It enables the agent to autonomously generate tasks, explore tool actions, and collect step-wise preference feedback via an AI verifier. The agent is then optimized using DPO based on this feedback. Experimental results on GTA and GAIA benchmarks show that SPORT significantly outperforms strong supervised and DPO baselines, and achieves competitive results with some closed-source models. The framework demonstrates that agents can improve tool usage capability purely through self-exploration and AI feedback, providing a scalable solution for complex multimodal tasks.

**Questions:**

1. While the SPORT framework uses DPO for preference tuning, is there a reason why fully online RL-based methods such as GRPO or PPO-style optimization were not adopted?
2. If I directly apply the SPORT framework or preference tuning on MAT-Qwen2-VL-7B, rather than just the base Qwen2-VL-7B, would the agent performance improve further? Has this variant been tried?

**Ethical Concerns:**

["NO or VERY MINOR ethics concerns only"]

**Final Justification:**

Most of my concerns are addressed. I will maintain my score.

**Limitations:**

yes

**Paper Formatting Concerns:**

No formatting concern.

**Quality:**

3

**Strengths And Weaknesses:**

Strengths:
1. Annotation free: SPORT does not rely on any human-annotated data or pre-collected trajectories, which addresses a major bottleneck in current multimodal agent research and greatly reduces data collection costs.
2. Novel online self-exploration: The method leverages an iterative, online self-exploration and preference optimization process, allowing the agent to adaptively discover effective strategies and improve performance through interaction with the environment.

Weaknesses:
1. The paper does not always specify the exact models used for each component (e.g., task synthesis, file generation, step verification, filtering). More explicit clarification on which LLM or VLM is used for each step, including possible ablation on model choices, would improve reproducibility and help understand the framework's sensitivity to model selection. Could the authors provide more detailed information on the models, prompts, and configurations for all stages of the pipeline?
2. No Comparison with True Online RL: Although they ablate DPO vs. static DPO, they never compare against an online RL baseline (e.g., PPO or GRPO) on the same framework. It is unclear whether SPORT is merely an intermediate between offline DPO and true online RL, or if incorporating conventional RL techniques could yield further gains. A direct comparison to PPO/GRPO baselines would clarify SPORT’s position on the spectrum from static preference tuning to fully online reinforcement learning.

---

> ### Author Rebuttal · Authors · 2025-07-31
>
> Thank you for the detailed and thoughtful review. We appreciate your recognition of SPORT's ability to train multimodal agents entirely without human-annotated data, as well as the novelty of our self-exploration and preference optimization framework. To simplify the discussion, we use **W** for each weakness, **Q** for each question, and **A** for our response. For clarity, all mentions of Qwen2-VL-7B and Qwen2.5-7B refer to their Instruct variants. We address your comments one by one below.
>
> > **W1.** The paper does not always specify the exact models used for each component (e.g., task synthesis, file generation, step verification, filtering). More explicit clarification on which LLM or VLM is used for each step, including possible ablation on model choices, would improve reproducibility and help understand the framework's sensitivity to model selection. Could the authors provide more detailed information on the models, prompts, and configurations for all stages of the pipeline?
>
> **A.** We will add the following details about model choices into the main manuscript.
>
> - *Task Synthesis* (including query generation, file generation and task revision&filtering)：Qwen2-VL-7B
> We chose Qwen2-VL-7B for task synthesis. To assess sensitivity to this choice, we conducted a human study (detailed in Appendix C.1), comparing task quality generated by Qwen2-VL-7B versus GPT-4o-mini on "Naturalness" and "Reasonableness". As shown in Table 7, both models produce tasks of comparable quality, demonstrating that Qwen2-VL-7B is robust and capable of high-quality task generation.
>
> **Table 7.** User study for open-source vs. closed-source models generated tasks. Scores are scaled from 1 to 10, and a higher score denotes better quality.
>
> | Model| Task Naturalness | Task Reasonableness |
> |-|-|-|
> | GPT-4o-mini|8.71|8.37|
> | Qwen2-VL-7B|8.75|8.35|
>
> - *Controller*: Tuned-Qwen2-VL-7B
> - *Step Verification*: Qwen2.5-7B
>
> We prioritized open-source models to ensure reproducibility and facilitate future research.  We selected Qwen2.5-7B as the verifier because qualitative analysis showed it performed better in evaluating code accuracy and tool selection appropriateness compared to Qwen2-VL-7B. We attribute this to Qwen2-VL-7B's image inputs occupying significant context window space, limiting available context for verifier instructions. Quantitative experiments confirm this: re-generating preference annotations for using Qwen2-VL-7B as verifier yielded 55.13% accuracy on the GTA benchmark, while Qwen2.5-7B achieved 60.26% (Table A), validating our choice. Additionally, our human evaluation (lines 324–330 and Table 5) shows that Qwen2.5-7B achieves an 82% alignment rate in step verification with human participants, confirming the effectiveness of our design choice.
>
>
> **Table A.** Ablation about AI verifier, accuracies (%) on GTA benchmark.
> | Baseline | Qwen2-VL-7B as Verifier | Qwen2.5-7B as Verifier |
> |-|-|-|
> | 53.85| 55.13 |60.26|
>
>
>
>
> - *Prompts and Configurations*: All system prompts are provided in Appendix E (Figures 6-14). Training configurations are detailed in Section 4.1. Code and data will be open-sourced for replication.
>
>
> > **W2.** No Comparison with True Online RL: Although they ablate DPO vs. static DPO, they never compare against an online RL baseline (e.g., PPO or GRPO) on the same framework. It is unclear whether SPORT is merely an intermediate between offline DPO and true online RL, or if incorporating conventional RL techniques could yield further gains. A direct comparison to PPO/GRPO baselines would clarify SPORT’s position on the spectrum from static preference tuning to fully online reinforcement learning.
>
> > **Q1.** While the SPORT framework uses DPO for preference tuning, is there a reason why fully online RL-based methods such as GRPO or PPO-style optimization were not adopted?
>
> **A.** Thank you for this excellent question. We have not adopted a fully online RL like PPO/GRPO for the following reasons.
>
> 1. Absence of reliable rewards. SPORT targets multimodal reasoning tasks *without ground-truth answers or labels*. On-policy methods like PPO/GRPO are sensitive to reward quality, and constructing reliable reward signals in this setting is non-trivial. Inaccurate rewards often lead to unstable or ineffective training.
>
> 2. Low trajectory success rates. In complex multimodal scenarios, most sampled trajectories fail, making it difficult to identify successful rollouts for policy updates in PPO/GRPO. SPORT's step-wise DPO approach better utilizes the correct steps within failed trajectories, improving data utilization.
>
> 3. High sampling cost and low efficiency. Each task requires multi-step interactions with real-world tools (e.g., web search, OCR, code execution), which are both computationally expensive and time-intensive. This results in very low sampling throughput, significantly slowing down the collection of trajectories needed for on-policy updates. As a result, applying PPO/GRPO becomes infeasible in our setting.
>
> Given these challenges around reward quality, task difficulty, and sampling efficiency, SPORT uses Step-wise DPO for practical and stable training. In future work, we plan to explore ways to address these sampling environment challenges, such as designing systems for more realistic, fast, and accurate rewards.
>
>
> >**Q2.** If I directly apply the SPORT framework or preference tuning on MAT-Qwen2-VL-7B, rather than just the base Qwen2-VL-7B, would the agent performance improve further? Has this variant been tried?
>
>
>
> **A.** Thank you for your suggestion. SPORT method already incorporates SFT on MAT data, as described in *lines 193–196* of the manuscript. To provide a more comprehensive analysis, we additionally applied SPORT directly to the base Qwen2-VL-7B without MAT.
>
> The results in Table B show that SPORT improves performance in both cases. Applying SPORT to the MAT baseline yields the best accuracy (60.26%), while applying it directly to the base Qwen2-VL-7B also results in a substantial gain (55.13% vs. 42.31%). The 5.13% additional improvement over MAT alone demonstrates that SPORT can effectively complement existing tuning methods, and can serve as a general enhancement strategy for already-trained agents. These results will be included in the revised manuscript.
>
> **Table B.** Answer accuracies(%) on the GTA benchmark.
> |Qwen2-VL-7B | MAT-Qwen2-VL-7B|SPORT-Qwen2-VL-7B|SPORT-MAT-Qwen2-VL-7B|
> |-|-|-|-|
> |42.31|53.85|55.13|60.26|

---

> > ### Comment · Reviewer_VNFD · 2025-08-04
> >
> > Thanks for authors' response. My concerns are addressed. I will maintain my positive score.

---

> > > ### Author Response · Authors · 2025-08-04
> > > **Thanks for your review!**
> > >
> > > Thank you so much for your thoughtful feedback, which helped us improve the paper. We're truly glad that your concerns have been addressed.

---

### Note · Authors · 2025-08-14

Dear Reviewers and Area Chair,

We sincerely thank all reviewers and the AC for their constructive feedback and positive engagement throughout the review process.

**Acknowledged Strengths**. We are encouraged that reviewers recognized our contributions:
- The annotation-free SPORT framework that eliminates dependency on expensive human annotations while achieving strong performance (`VNFD`, `avvW`, `Zye7`).
- Novel online self-exploration with step-level preference optimization that enables learning from partial successes (`VNFD`, `Zye7`).
- Strong empirical results demonstrating clear improvements over baselines and competitive performance with closed-source model powered agents (`avvW`, `rUdq`).
- Practical design working with open-source models and generating a valuable 16K preference dataset for the community (`rUdq`, `Zye7`).

**Key Concerns Addressed During Rebuttal.**
- Controller/verifier model design choice: We provided a clear rationale for our controller and verifier on model selection, explaining the design trade-offs and presenting ablations to justify our choices.
- Efficiency and cost analysis: We systematically analyzed and compared the time and computational cost of the SPORT framework to provide a transparent view of its efficiency.
- Comparison to alternatives: We discussed the applicability of different training paradigms in our setting and showed how iterative self-exploration serves as an effective solution.
- Robustness and sensitivity studies: We evaluated SPORT’s robustness to variations in task generation and dataset size, confirming stable performance across diverse conditions.

**Planned Revision for Manuscript.**
- Incorporate our insights and discussions on model choice into the manuscript.
- Include our efficiency and time cost analyses to provide a comprehensive view of efficiency.
- Add the experiments and ablations from rebuttal, including verifier choice analyses, task diversity studies, and scaling experiments with different dataset sizes.
- Release all code, data, and prompts to allow full reproduction of the results (as mentioned in the paper).

We are grateful that all reviewers confirmed maintaining their original positive ratings (`Zye7`:5, `VNFD`:4, `rUdq`:4, `avvW`:4) and appreciate the constructive discussions throughout the review. We look forward to incorporating these valuable insights into the final version. Thank you again for the valuable feedback that greatly strengthened this work.

Best,

Authors

---

### Decision · Program_Chairs · 2025-09-17

**Decision:**

Accept (poster)

**Comment:**

This paper introduces SPORT, an annotation-free, iterative self-exploration framework for training multimodal tool-use agents. The method mitigates the reliance on costly human annotations by generating tasks, sampling step-level actions, and leveraging an AI verifier with step-wise preference optimization (DPO). Experiments on GTA and GAIA benchmarks show consistent improvements over strong supervised and DPO baselines, with competitive results compared to closed-source agents. In addition, the authors also contributed a 16K preference data for the research community.

This submission has several strengths including: 1) Annotation-free training pipeline which does not rely on human annotations but a self-evolvement system to gradually acquire the knowledge to interact with the environment. 2) Novel training paradigm where step-level preference optimization from self-generated tasks and verifier feedback enables learning from partial successes and 3) Strong experimental results showcasing the compared with SFT baselines and other related works. Reviewers also posed a few concerns about the method including the technical novelty of the proposed method and the analysis on the effect of chosen models as verifier and controller. Delightfully, the authors spent tremendous efforts to address the concerns raised by the reviewers with more experimental results and analysis. After the rebuttal, the reviewers converged on positive assessments: one accept (5) and three borderline accepts (4), all maintaining their scores after rebuttal.

The AC finds that the paper presents a technically solid, practically valuable contribution, with sufficient novelty at the level of training paradigm. While some limitations remain (performance gap to large closed-source models, limited architectural innovations), the strengths outweigh the concerns. As such, the ACs recommend a clear acceptance.